# Test-time Alignment of Diffusion Models without Reward Over-optimization

**Sunwoo Kim**[1][*]  **Minkyu Kim**[2]  **Dongmin Park**[2][†]
[1] Seoul National University   [2] KRAFTON
ksunw0209@snu.ac.kr  {minkyu.kim, dongmin.park}@krafton.com

## Abstract

Diffusion models excel in generative tasks, but aligning them with specific objectives while maintaining their versatility remains challenging. Existing fine-tuning methods often suffer from reward over-optimization, while approximate guidance approaches fail to optimize target rewards effectively. Addressing these limitations, we propose a training-free, test-time method based on Sequential Monte Carlo (SMC) to sample from the reward-aligned target distribution. Our approach, tailored for diffusion sampling and incorporating tempering techniques, achieves comparable or superior target rewards to fine-tuning methods while preserving diversity and cross-reward generalization. We demonstrate its effectiveness in single-reward optimization, multi-objective scenarios, and online black-box optimization. This work offers a robust solution for aligning diffusion models with diverse downstream objectives without compromising their general capabilities. Code is available at https://github.com/krafton-ai/DAS.

## 1 Introduction

Diffusion models (Sohl-Dickstein et al., 2015; Ho et al., 2020; Song et al., 2021c; Park et al., 2024) have revolutionized generative AI, excelling in tasks from Text-to-Image (T2I) generation (Rombach et al., 2022) to protein structure design (Watson et al., 2023). However, diffusion models are typically pre-trained on large uncurated datasets that may not accurately represent the desired target distribution. For instance, in the T2I generation, real users want to produce aesthetically pleasing images while faithful to prompt instructions, rather than generating random internet images from the pre-trained dataset. Also, one might want to produce only specific cartoon character images, rather than general styles. These challenges underscore the importance of *alignment*, a process to adapt diffusion models for specific customized rewards.

Existing alignment approaches mainly fall into two categories: (1) fine-tuning and (2) guidance methods. Fine-tuning approaches, including Reinforcement Learning (RL) (Fan et al., 2024; Black et al., 2023) and direct backpropagation (Clark et al., 2024; Prabhudesai et al., 2024), have shown promising results in optimizing target rewards. However, it often suffers from the reward *over-optimization* problem, sacrificing general image quality and diversity (Clark et al., 2024; Gao et al., 2022). On the other hand, guidance methods (Bansal et al., 2023; Yu et al., 2023; Song et al., 2023; He et al., 2024) offer a training-free alternative that stays closer to the pre-trained model distribution. Meanwhile, they suffer from the reward *under-optimization* problem, failing to effectively optimize target rewards due to relying on estimated inference-time corrections of the generation process.

To address these limitations, we propose **D**iffusion **A**lignment as **S**ampling (**DAS**), a *test-time* algorithm that both achieves effective reward alignment and preserves model generalization. Our approach leverages Sequential Monte Carlo (SMC) sampling (Del Moral et al., 2006) to enable high-reward diffusion guidance by evaluating multiple possible solutions in parallel and selecting the best candidates during the generation process. By carefully designing intermediate target distributions with tempering techniques, DAS achieves high sample efficiency in terms of required candidates, as we demonstrate both theoretically and empirically.

---

[*]This work was done during an internship at KRAFTON.
[†]Corresponding author.

To validate its effectiveness for optimizing target reward without over-optimization, we apply DAS to Stable Diffusion v1.5 (Rombach et al., 2022), targeting aesthetic reward, e.g., LAION aesthetic score (Schuhmann, 2022) and human preference, e.g., PickScore (Kirstain et al., 2023). Without the computational burden of training or extensive hyperparameter tuning, DAS outperforms all fine-tuning baselines in two target scores, while not sacrificing cross-reward generalization and output diversity. This superiority is consistent with a more recent diffusion backbone, SDXL (Podell et al., 2024), and more complex reward functions such as CLIPScore (Hessel et al., 2021) and Compressibility (Black et al., 2023). We further demonstrate the efficacy of DAS in multi-objective optimization, achieving a new Pareto front when jointly optimizing CLIPScore (Hessel et al., 2021) and aesthetic score. Moreover, the diverse sampling ability is especially beneficial in online settings with limited reward queries. In such a difficult scenario, while existing methods drop the scores of unseen rewards due to severe over-optimization, DAS improves the pre-trained T2I model by up to 20% in both target and unseen rewards.

In summary, our main contributions are:

- We propose DAS, a test-time method for aligning diffusion models with arbitrary rewards while preserving general capabilities.
- We provide theoretical analysis of DAS's asymptotic properties, proving the benefits of tempering in SMC sampling for diffusion models.
- We empirically validate DAS's effectiveness across diverse scenarios, including single-reward, multi-objective, and online black-box optimization tasks.

## 2  RELATED WORK

### 2.1  FINE-TUNING DIFFUSION MODELS FOR ALIGNMENT

Aligning pretrained models through fine-tuning has been extensively studied in language models (Ziegler et al., 2020; Ouyang et al., 2022; Rafailov et al., 2023). For diffusion models, several approaches have emerged. Lee et al. (2023) and Wu et al. (2023b) employ supervised fine-tuning with preference-based reward models. Black et al. (2023) and Fan et al. (2024) formulate sampling as a Markov decision process and apply reinforcement learning (RL) to maximize rewards. Xu et al. (2024); Clark et al. (2024) and Prabhudesai et al. (2024) fine-tune by direct backpropagation through differentiable reward models. These approaches, however, face challenges with reward over-optimization (Gao et al., 2022; Coste et al., 2024), which may distort alignment or reduce sample diversity. KL regularization has been proposed as a mitigation strategy (Fan et al., 2024; Uehara et al., 2024a), inspired by its success in language models (Stiennon et al., 2020; Ouyang et al., 2022; Korbak et al., 2022). Section 3.2 examines the limitations of this approach, focusing on the mode-seeking behavior observed in the context of variational inference. While diffusion-based samplers (Zhang & Chen, 2022; Vargas et al., 2023; Berner et al., 2024; Sanokowski et al., 2024) use similar training objective to sample from multimodal, unnormalized target density, the fine-tuning setup makes training more susceptible to mode collapse (Appendix F). Alternatively, Zhang et al. (2024) approached over-optimization in RL fine-tuning through inductive and primacy biases.

### 2.2  GUIDANCE METHODS

Building on the score-based formulation of diffusion models (Song et al., 2021c), various guidance methods have been developed. While classifier guidance (Dhariwal & Nichol, 2021) requires additional training, recent works approximate guidance to use off-the-shelf classifiers or reward models directly (Ho et al., 2022; Song et al., 2022; Chung et al., 2023; Bansal et al., 2023; Yu et al., 2023; Song et al., 2023; Yoon et al., 2023; He et al., 2024). These methods rely on Tweedie's formula (Efron, 2011; Chung et al., 2023) for prediction of the original data given noisy data, but our experiments indicate that such inaccurate prediction limits effectiveness in maximizing complex rewards. In contrast, SMC methods have been applied to address inexactness (Trippe et al., 2023; Cardoso et al., 2024; Wu et al., 2023a; Dou & Song, 2024) or bypass the need of calculating guidance (Li et al., 2024). While SMC methods offer asymptotic exactness, naive applications may fail to sample from complex targets within finite samples due to inefficiency. Our approach incorporates a tempered SMC sampler to enhance sample efficiency, achieving comparable or superior performance to fine-tuning methods without additional training.

# 3 DIFFUSION ALIGNMENT AS SAMPLING (DAS)

This section formulates the diffusion alignment problem as sampling from a reward-aligned distribution, examines limitations of existing methods, and introduces DAS, a Sequential Monte Carlo (SMC) based algorithm with theoretical guarantees for asymptotic exactness and sample efficiency.

## 3.1 PROBLEM SETUP: ALIGNING DIFFUSION MODELS WITH REWARDS

Aligning diffusion models with rewards can be seen as finding a new distribution that maximizes the expectation given reward $r$. Formally, it can be written as solving:

$$p_{\text{tar}} = \arg\max_p \mathbb{E}_{x \sim p}[r(x)]. \tag{1}$$

However, this approach may lead to reward over-optimization (Gao et al., 2022), disregarding the pre-trained distribution. To mitigate this, we employ KL regularization (Uehara et al., 2024a):

$$p_{\text{tar}} = \arg\max_p \mathbb{E}_{x \sim p}[r(x)] - \alpha D_{\text{KL}}(p \| p_{\text{pre}}) \tag{2}$$

where $p_{\text{pre}}$ is the sample distribution of the pre-trained diffusion model. Following Rafailov et al. (2023), it is straightforward to show that the target distribution can be written in an equivalent form:

$$p_{\text{tar}}(x) = \frac{1}{\mathcal{Z}} p_{\text{pre}}(x) \exp\left(\frac{r(x)}{\alpha}\right) \tag{3}$$

where $\mathcal{Z}$ is normalization constant. We frame the diffusion alignment problem as sampling from this reward-aligned target distribution $p_{tar}$. Note, however, that we only have access to an unnormalized density of $p_{tar}$ and its evaluation requires running a probability flow ODE (Song et al., 2021c), even for a single sample, making the sampling problem highly non-trivial.

Before we continue, we introduce binary optimality variable $\mathcal{O} \in \{0, 1\}$ with $p(\mathcal{O} = 1|x) \propto \exp(r(x)/\alpha)$, where samples with high reward are interpreted as more likely *optimal*. Then the posterior $p(x|\mathcal{O} = 1)$ characterizes the distribution of samples that achieve high rewards. Using Bayes' rule with prior $p = p_{\text{pre}}$ give $p(x|\mathcal{O} = 1) \propto p(x)p(\mathcal{O} = 1|x) = p_{\text{pre}}(x) \exp(r(x)/\alpha) \propto p_{\text{tar}}(x)$, revealing the equivalence between two perspectives. We omit '=1' hereafter following convention.

## 3.2 LIMITATIONS OF EXISTING METHODS

Previous approaches to sampling from the target distribution (Equation 3) primarily fall into two categories: fine-tuning and direct sampling using approximate guidance. Here, we first demonstrate how these approaches struggle to sample from multimodal target distributions, even for simple Gaussian mixtures, and explain their limitations leading to potential failures.

Figure 1 illustrates the failure modes of two approaches. Fine-tuning methods (RL, direct backpropagation) fail to fit all modes of the multimodal target distribution $p_{\text{tar}}$, depicting their mode-seeking behavior. Approximate guidance results in low rewards, failing to effectively optimize target reward, portraying the inexactness of the guidance. In contrast, our SMC-based method successfully samples from all modes, achieving the lowest Earth Mover's Distance between $p_{\text{tar}}$ with high reward.

We first investigate the source of the *mode-seeking* behavior of fine-tuning methods. Fine-tuning methods can be seen as variational inference, with the following objective (Rafailov et al. (2023)):

$$\underset{\theta}{\text{minimize}}\, D_{\text{KL}}(p_\theta \| p_{\text{tar}}). \tag{4}$$

This can be optimized using reinforcement learning (RL) (Fan et al., 2024) or direct backpropagation (Uehara et al., 2024a). However, the mode-seeking behavior of reverse KL divergence (Chan et al., 2022; Wang et al., 2023) may cause the model to fit only the modes of the target distribution, especially when $p_{tar}$ is multimodal (See Figure 1). This connects to low diversity of fine-tuning methods, which we further demonstrate in Section 4 for real-world examples.

Next, we turn to approximate guidance methods. If the exact score function of the posteriors

$$\nabla_{x_t} \log p_t(x_t|\mathcal{O}) = \nabla_{x_t} \log p_t(x_t) + \nabla_{x_t} \log p(\mathcal{O}|x_t), \tag{5}$$

is known, one can use reverse diffusion for generation (Song et al., 2021c), where marginal $p_t(x_t)$ and conditional distribution $p_t(x_t|\mathcal{O})$ is defined by the forward diffusion process. However, to

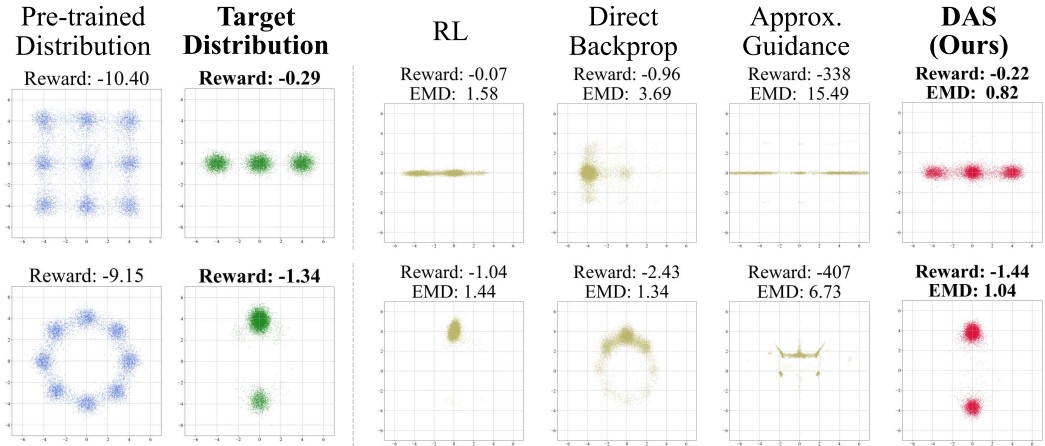

Figure 1: **SMC method excels in sampling from the target distribution compared to existing approaches.** Left of dashed line: Samples from pre-trained model trained on mixture of Gaussians, reward-aligned target distribution $p_{tar}$. Right of dashed line: methods for sampling from $p_{tar}$ including previous methods (RL, direct backpropagation, approximate guidance) and ours using SMC. Top: reward $r(X, Y) = -X^2/100 - Y^2$, bottom: reward $r(X, Y) = -X^2 - (Y-1)^2/10$. EMD denotes sample estimation of Earth Mover's Distance. Our SMC-based method outperforms existing approaches in capturing multimodal target distributions, as evidenced by lower EMD and successful sampling from all modes. Note that samples may exist outside the grid.

sidestep the intractable integration $p(\mathcal{O}|x_t) = \int p(\mathcal{O}|x_0)p(x_0|x_t)\,dx_0$, line of works (Chung et al., 2023; Yu et al., 2023; Bansal et al., 2023; Song et al., 2023) rely on the approximation $p(\mathcal{O}|x_t) = \mathbb{E}_{p(x_0|x_t)}[p(\mathcal{O}|x_0)|x_t] \approx p(\mathcal{O}|\hat{x}_0(x_t))$ where $\hat{x}_0 := \mathbb{E}[x_0|x_t]$ is given by the Tweedie's formula (Efron, 2011; Chung et al., 2023). Finally, replacing $p(\mathcal{O}|x_0) \propto \exp\left(r(x_0)/\alpha\right)$, approximate guidance is given as

$$\nabla_{x_t} \log p(\mathcal{O}|x_t) \approx \frac{1}{\alpha}\nabla_{x_t}\hat{r}(x_t), \qquad (6)$$

where $\hat{r}(\cdot) := r(\hat{x}_0(\cdot))$. However, predicting clean data with noise causes errors, especially early in sampling when the noise is large, making it difficult to sample exactly from $p_{\text{tar}}$ (He et al., 2024).

### 3.3 Sampling from Reward-aligned Target Distribution via Tempered SMC

Fine-tuning often leads to over-optimization, while approximate guidance methods struggle with reward optimization. To improve approximate guidance, we can use multiple candidate latents (particles) during sampling, selecting those with high predicted rewards that stay close to the pre-trained diffusion model's distribution. This approach leverages Sequential Monte Carlo (SMC) methods, which take incremental guided steps rather than sampling directly from the target distribution. At each diffusion step, the process evaluates and resamples candidates based on both reward scores and alignment with the pre-trained model, ultimately producing samples that satisfy both criteria.

Traditional SMC typically requires thousands of particles, making it computationally expensive for diffusion models. However, by employing techniques like tempering, we achieve high-quality, reward-aligned samples with fewer particles, making the method practical for real-world applications. To formally describe our approach, we first outline the key design choices of SMC samplers that enable this guided sampling process:

- Sequence of intermediate target distributions $\pi_t(x_t) = \tilde{\gamma}_t(x_t)/\mathcal{Z}_t$ for $t = 0 : T$ that bridge between the prior $\pi_T$ and target distribution $\pi_0$, where $\gamma_t$ is unnormalized density of $\pi_t$.

- Backward kernels [1] $L_t(x_t|x_{t-1})$ which define intermediate joint distributions

$$\bar{\pi}_t(x_{t:T}) := \pi_t(x_t) \prod_{s=t+1}^{T} L_s(x_s|x_{s-1}). \qquad (7)$$

---

[1]Backward respect to sampling procedure, which is the same time direction with a forward diffusion process.

- Proposals, or transition kernels $m_{t-1}(x_{t-1}|x_t)$ for sequential sampling.
- Weights

$$w_{t-1}(x_{t-1}, x_t) := \frac{\bar{\pi}_{t-1}(x_{t-1:T})}{\bar{\pi}_t(x_{t:T})m_{t-1}(x_{t-1}|x_t)} \propto \frac{\tilde{\gamma}_{t-1}(x_{t-1})}{\tilde{\gamma}_t(x_t)} \frac{L_t(x_t|x_{t-1})}{m_{t-1}(x_{t-1}|x_t)}, \tag{8}$$

where proposed particles are resampled from $\text{Multinomial}(x_{t-1}^{1:N}; w_{t-1}^{1:N})$.

A more detailed and theoretical introduction to SMC can be found in Appendix B.

### 3.3.1 BACKWARD KERNEL

To incorporate pre-trained diffusion models, we define the backward kernel using Bayes' rule with general stochastic diffusion samplers. For any stochastic diffusion sampler $p_\theta(x_{t-1}|x_t) = \mathcal{N}(\mu_\theta(x_t, t), \sigma_t^2 I)$, we define the backward kernels as:

$$L_t(x_t|x_{t-1}) := \frac{p_\theta(x_{t-1}|x_t)p_t(x_t)}{p_{t-1}(x_{t-1})}. \tag{9}$$

This formulation also serves as an approximation for general non-Markovian forward processes given pre-trained generation processes (Song et al., 2021a).

### 3.3.2 INTERMEDIATE TARGETS: APPROXIMATE POSTERIOR WITH TEMPERING

As stated in section 3.2, sampling from the target $p(x_0|\mathcal{O})$ requires score functions of the true posteriors $p_t(x_t|\mathcal{O})$. Instead, approximate guidance gives a score function of an alternative distribution, which we refer to as the approximate posterior:

$$\hat{p}_t(x_t|\mathcal{O}) \propto p_t(x_t)p(\mathcal{O}|\hat{x}_0(x_t)) \propto p_t(x_t)\exp\left(\frac{\hat{r}(x_t)}{\alpha}\right). \tag{10}$$

However, we can't sample even from these approximate posteriors since they are not defined by any forward diffusion process anymore. Nevertheless, this approximate posterior becomes exact at $t = 0$ as $\hat{x}_0 = x_0$, thus defining a sequence of distributions interpolating $p_T = \mathcal{N}(0, \sigma_T^2 I)$ and $p_{\text{tar}}$ which can be incorporated as intermediate targets for SMC sampler. Since prediction $\hat{x}_0$ gets more accurate as $t$ goes to 0, the approximate posteriors get closer to the true posteriors while the error may be large at the beginning of sampling. Hence, we add tempering for intermediate targets as:

$$\pi_t(x_t) \propto p_t(x_t)p(\mathcal{O}|\hat{x}_0(x_t))^{\lambda_t} \propto p_t(x_t)\exp\left(\frac{\lambda_t}{\alpha}\hat{r}(x_t)\right) =: \tilde{\gamma}_t(x_t), \tag{11}$$

which can interpolate $\pi_T = p_T$ to $\pi_0 = p_{\text{tar}}$ more smoothly where $0 = \lambda_T \leq \lambda_{T-1} \leq \cdots \leq \lambda_0 = 1$ is sequence of inverse temperature parameters.

While modern SMC samplers often use adaptive tempering (Chopin & Papaspiliopoulos, 2020; Murphy, 2023), we find out simply setting $\lambda_t = (1 + \gamma)^t - 1$ works well in our setting where $\gamma$ is a hyperparameter. In Section 4.1, we compare different tempering schemes and explain how to select $\gamma$. To the best of our knowledge, this adaptation of density tempering is novel among works applying SMC methods to diffusion sampling.

### 3.3.3 PROPOSAL: APPROXIMATING LOCALLY OPTIMAL PROPOSAL

Given the backward kernels and intermediate targets, we derive the locally optimal proposal that minimizes the variance of the weights. Minimizing weight variance ensures more uniform importance among particles, thereby enhancing sample efficiency.

**Proposition 1** (Locally Optimal Proposal). *The locally optimal proposal $m_{t-1}^*(x_{t-1}|x_t)$ that minimizes the conditional variance $\text{Var}(w_{t-1}(x_{t-1}, x_t)|x_t)$ is given by*

$$m_{t-1}^*(x_{t-1}|x_t) \propto \exp\left(-\frac{1}{2\sigma_t^2}\|x_{t-1} - \mu_\theta(x_t, t)\|^2 + \frac{\lambda_{t-1}}{\alpha}\hat{r}(x_{t-1})\right). \tag{12}$$

*proof. The full proof can be found in Appendix C.1.1*

Since sampling from $m^*$ is non-trivial, we adapt Gaussian approximation of $m^*$ as our proposal:

$$m_{t-1}(x_{t-1}|x_t) := \mathcal{N}\left(\mu_\theta(x_t, t) + \sigma_t^2 \frac{\lambda_{t-1}}{\alpha}\nabla_{x_t}\hat{r}(x_t), \sigma_t^2 I\right),\quad (13)$$

where we used first-order Taylor approximation $r(\hat{x}_0(x_{t-1})) \approx r(\hat{x}_0(x_t)) + \langle\nabla_{x_t}r(\hat{x}_0(x_t)), x_{t-1} - x_t\rangle$ and $\hat{x}_0(\cdot, t) \approx \hat{x}_0(\cdot, t-1)$ [2].

Tempering further improves this approximation by reducing errors from linear approximation, thereby decreasing weight variance. It also mitigates off-manifold guidance (He et al., 2024), particularly in early sampling stages. As $\lambda_t$ increases from 0 to 1, it gradually guides sampling towards the target while minimizing weight degeneracy and manifold deviation. Section 4 provides empirical validation of these effects.

Finally, the unnormalized weights for each particle are calculated as

$$w_{t-1}(x_{t-1}, x_t) = \frac{\tilde{\gamma}_{t-1}(x_{t-1})}{\tilde{\gamma}_t(x_t)}\frac{L_t(x_t|x_{t-1})}{m_{t-1}(x_{t-1}|x_t)} = \frac{p_\theta(x_{t-1}|x_t)\exp\left(\frac{\lambda_{t-1}}{\alpha}\hat{r}(x_{t-1})\right)}{m_{t-1}(x_{t-1}|x_t)\exp\left(\frac{\lambda_t}{\alpha}\hat{r}(x_t)\right)},\quad (14)$$

for $t = 1:T$ and $w_T(x_T) = \exp\left(\frac{\lambda_T}{\alpha}\hat{r}(x_T)\right)$ which are used for resampling. The pseudo-code of the final algorithm with adaptive resampling is given in Algorithm A.1.

### 3.3.4 ASYMPTOTIC BEHAVIOR

This section presents asymptotic analysis results for DAS. We first demonstrate asymptotic exactness, a key property distinguishing SMC methods from other approximate guidance approaches.

**Proposition 2** (Asymptotic Exactness). *(Informal) Under regularity conditions, sample estimation of $\mathbb{E}_{X \sim p_{tar}}[\varphi(X)]$ given by DAS converge to the true expectation almost surely for test functions $\varphi$.*

Although SMC samplers are asymptotically exact, their sample efficiency depends on design choices. Using a Central Limit Theorem analysis, we bound the asymptotic variance of sample estimations. This approach allows us to prove the benefits of tempering for sample efficiency, providing theoretical justification beyond intuitive advantages.

**Proposition 3** (Asymptotic Variance and Sample Efficiency). *(Informal) Under the same regularity conditions as Proposition 2, the upper bound of asymptotic variances of sample estimations given by DAS when tempering is used, i.e. $\lambda_t$'s are not all 1 for $t = 0:T$, are always smaller or equal to when tempering isn't used, i.e. $\lambda_t$'s are all 1 for $t = 0:T$.*

These propositions further imply setwise convergence of empirical measures to $p_{tar}$ and quantify the asymptotic error, which is reduced with tempering. Formal statements and proofs are provided in Appendix C.2.

## 4 EXPERIMENTS

The main benefits of DAS are twofold: (1) it can avoid over-optimization by directly sampling from the target distribution, and (2) it is efficient since there is no need for additional training. We investigate these benefits through various experiments by addressing the following questions:

- Can DAS effectively optimize a single reward while avoiding over-optimization? (§4.1)
- Can DAS optimize multiple rewards all at once without training for each combination? (§4.2)
- Can DAS effectively search diverse viable solutions in an online black-box optimization? (§4.3)
- Does tempering increase sample efficiency as predicted by the theory? (§4.1)

### 4.1 SINGLE REWARD

#### 4.1.1 EXPERIMENT SETUP

**Tasks.** For single reward tasks, we use aesthetic scores (Schuhmann et al., 2022) and human preference evaluated by PickScore (Kirstain et al., 2023) as objectives. For fine-tuning methods, we used

---

[2] $\hat{x}_0$ use noise prediction of pre-trained diffusion model, in which the output depends on time.

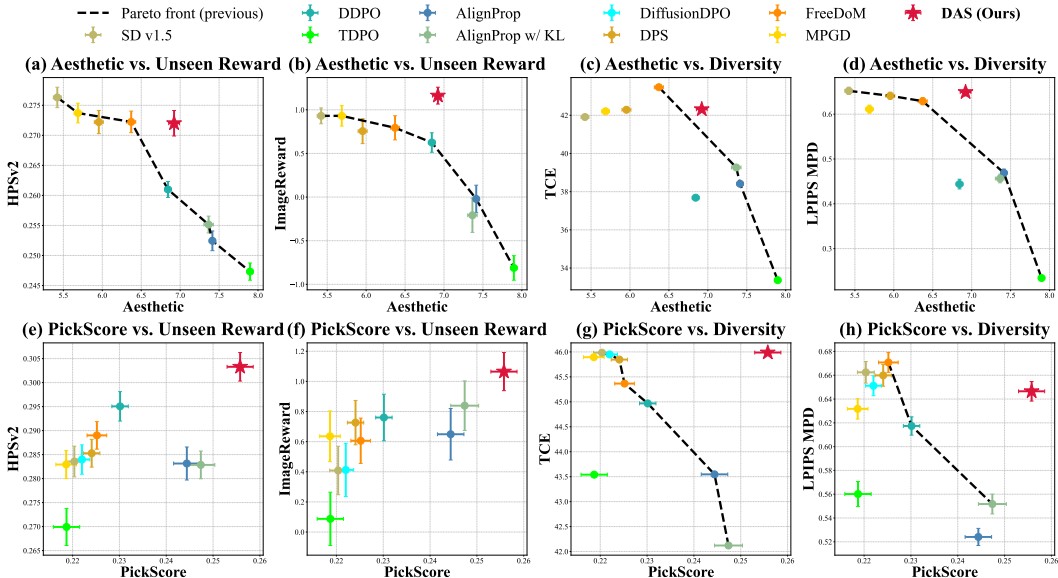

Figure 2: **Target Reward vs. Evaluation Metrics.** Top: target is the aesthetic score, bottom: target is PickScore. (a), (e) and (b), (f): evaluation of cross-reward generalization using HPSv2 and ImageReward, respectively. (c), (g) and (d), (h): evaluation of diversity using Truncated CLIP Entropy (TCE) and mean pairwise distance (MPD) calculated with LPIPS, respectively. Our method reach similar or better target reward compared to fine-tuning methods (DDPO, AlignProp) while maintaining cross-reward generalization and diversity like guidance methods (DPS, FreeDoM, MPGD), breaking through the Pareto-front of previous methods.

animals from Imagenet Deng et al. (2009) and prompts from Human Preference Dataset v2 (HPDv2) (Wu et al., 2023b) when training on aesthetic score and PickScore respectively, like previous settings (Black et al., 2023; Clark et al., 2024). Evaluation uses unseen prompts from the same dataset.

**Evaluation metrics.** We assess three aspects: target rewards, cross-reward generalization, and sample diversity. For cross-reward generalization, we use HPSv2 (Wu et al., 2023b) and ImageReward (Xu et al., 2024), both alternative rewards that measure human preference. For sample diversity, we use Truncated CLIP Entropy (TCE) (Ibarrola & Grace, 2024) which measures entropy of CLIP embeddings, and mean pairwise distance (MPD) calculated with LPIPS (Zhang et al., 2018) which quantifies perceptual differences.

**Baselines.** We employ Stable Diffusion v1.5 (Rombach et al., 2022) as the pre-trained model. Other baselines include fine-tuning methods (DDPO (Black et al., 2023), AlignProp (Prabhudesai et al., 2024), AlignProp with KL regularization, TDPO (Zhang et al., 2024), and DiffusionDPO (Wallace et al., 2024) for PickScore) and training-free guidance methods (DPS (Chung et al., 2023), FreeDoM (Yu et al., 2023), and MPGD (He et al., 2024)).

### 4.1.2 RESULTS

**Quantitative evaluation.** Figure 2 shows quantitative results on both the target reward and evaluation metrics. Fine-tuning methods generally cluster in the bottom right, indicating reward over-optimization with high target rewards but low diversity and poor generalization to similar rewards. AlignProp with KL exhibits a similar trend, failing to mitigate over-optimization due to mode-seeking behavior, as demonstrated in the mixture of Gaussian example (Section 3.2). TDPO, proposed as an alternative to early stopping and KL regularization, fails to effectively mitigate over-optimization for aesthetic scores and tends to under-optimize for PickScore. Conversely, guidance methods typically occupy the upper left quadrant, failing to optimize target rewards effectively. DAS consistently occupies the upper right quadrant, achieving high target rewards while maintaining cross-reward generalization and diversity, thus effectively mitigating over-optimization.

**Preserving diversity while optimizing rewards.** Figure 3 showcases samples generated from the prompt "crocodile," aimed at maximizing aesthetic score. Our approach demonstrates superior aes-

| Color | Count | Composition | Location | Style | Unusual |
|-------|-------|-------------|----------|-------|---------|

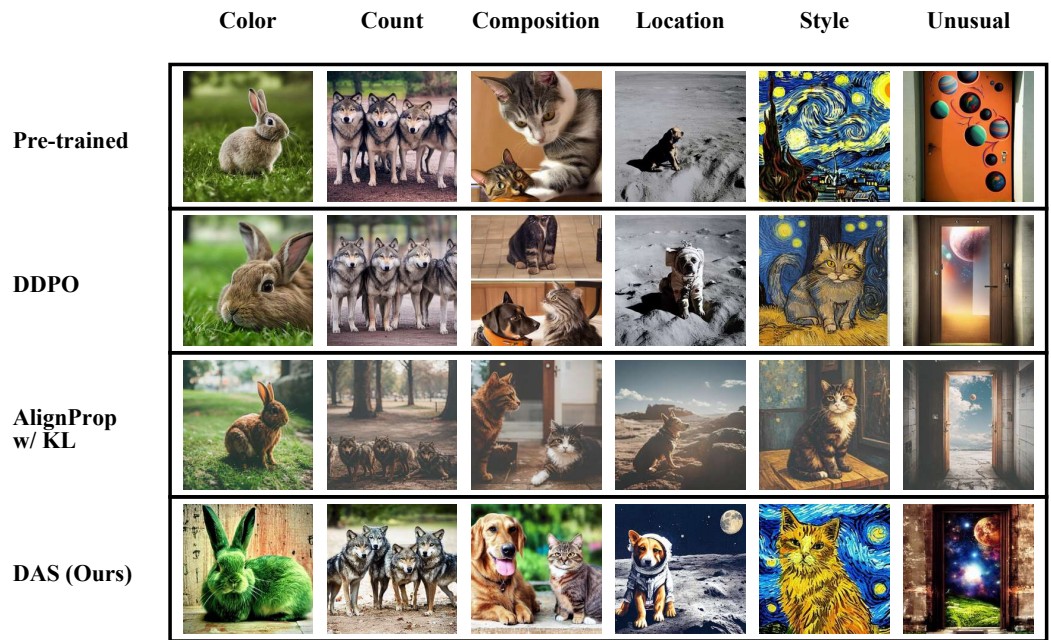

Figure 4: **Qualitative comparison of T2I alignment.** Target reward: PickScore. Unseen prompts: "*A green colored rabbit*" (color), "*Four wolves in the park*" (count), "*cat and a dog*" (composition), "*A dog on the moon*" (location), "*A cat in the style of Van Gogh's Starry Night*" (style), "*A door that leads to outer space*" (unusual). Samples generated by DAS used only 4 particles.

thetic appeal while preserving sample diversity and pre-trained features of the animal. In contrast, samples from fine-tuning methods deviate significantly from the pre-trained model's output and exhibit less diversity in colors, backgrounds, and appearances, indicating reward over-optimization.

**Improving T2I alignment.** Notably, in Figure 2, DAS substantially outperforms fine-tuning methods for the PickScore task across all metrics. To check whether the quantitative results align with actual human preferences, Figure 4 visualizes samples targeted to maximize PickScore across six categories: color, count, composition, location, style, and generating unusual scenes. DAS successfully generates aligned images with high visual appeal, even compared to fine-tuning baselines, thus effectively aligning the samples with human preferences. We provide additional results in Appendix G .

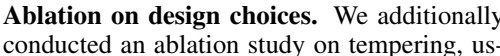

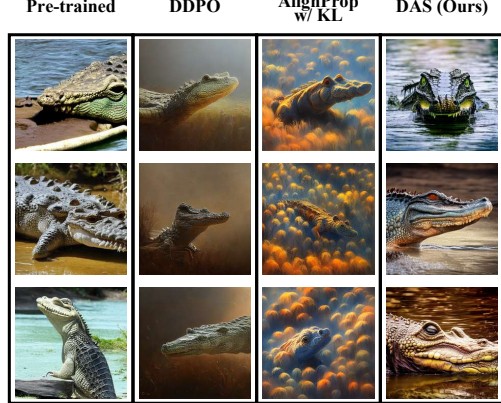

Figure 3: **Qualitative comparison of over-optimization and diversity.**

**Ablation on design choices.** We additionally conducted an ablation study on tempering, using the aesthetic score as target reward and ImageReward as evaluation for T2I alignment and cross-reward generalization. We also tested various tempering strategies, including $\gamma = 0.008$, $\gamma = 0.024$ (where $\lambda_t$ reaches 1 after 90 or 30 steps, respectively), and adaptive tempering (A.2). In Figure 5, without tempering, SMC suffers from over-optimization, even with 32 particles, resulting in low ImageReward. In contrast, with tempering, using only 4 or 8 particles can achieve both high aesthetic scores and ImageReward, greatly improving efficiency regardless of tempering schemes. Tempering also reduces deviation from the latent manifold (He et al., 2024), indicating fewer generations of OOD samples. These findings align with the theoretical predictions in Section 3.3.3 and 3.3.4. While the performance of DAS is robust to tempering schemes, we recommend low $\gamma$ with a tuned $\alpha$ for the best balance of quality and efficiency in general. Test-time scaling experiment in Appendix E further demonstrates that DAS can effectively leverage additional computation during inference.

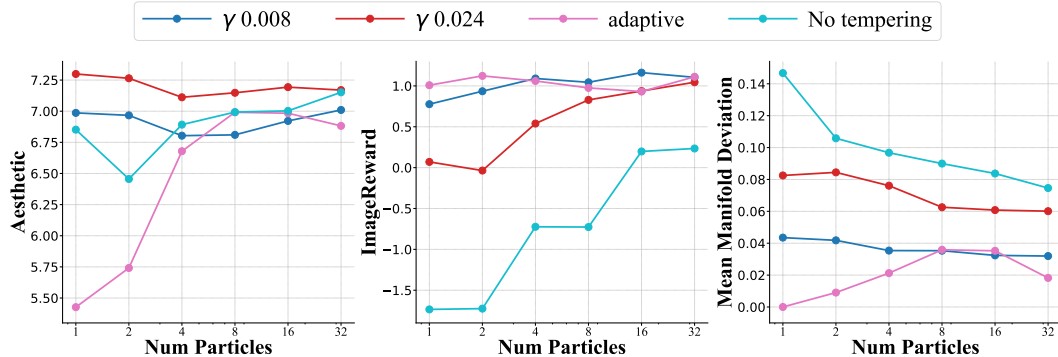

Figure 5: **Effect of tempering on sample efficiency and manifold deviation.** We compare different tempering schemes while changing the numbers of particles. (*left*) Number of Particles vs. aesthetic score (target reward), (*middle*) Number of Particles vs. ImageReward (unseen reward), (*right*) Number of Particles vs. mean deviation from latent manifold.

## 4.2 MULTI REWARDS

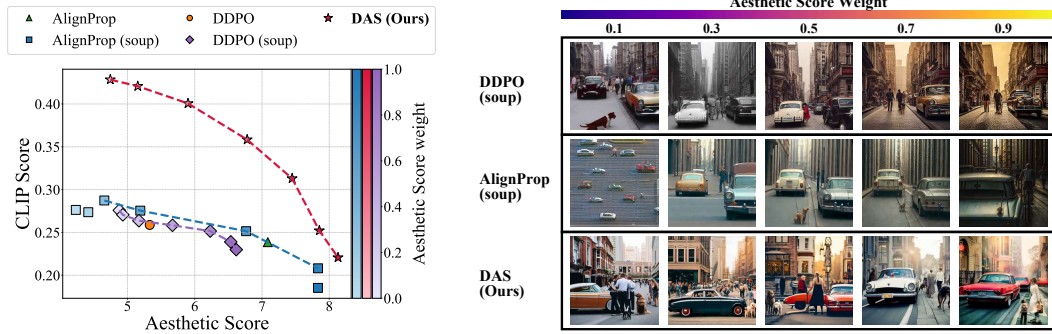

(a) Trade-off in multi-objective optimization.

(b) Generated samples according to reward weights

Figure 6: **Comparison of multi-objective optimization.** (a) DAS achieves a Pareto-optimal trade-off between the two rewards. (b) Prompt: "classic cars on a city street with people and a dog". Interestingly, DAS consistently generates 'dog' in the prompt when the aesthetic score weight gets lower than 0.9, while baselines fail to generate the dog for some images.

**Experiment setup.** Multi-objective optimization is crucial for real-world applications that balance competing goals (Deb et al., 2016) - for instance, generating visually appealing images while being faithful to prompts. To evaluate DAS in this practical setting, we combine aesthetic score and CLIPScore (Hessel et al., 2021), which measures image-text alignment. We use a weighted sum:

$$w \cdot \text{Aesthetic Score} + (1 - w) \cdot 20 \cdot \text{CLIPScore} \tag{15}$$

with $w \in \{0, 0.1, 0.3, 0.5, 0.7, 0.9, 1.0\}$. Baselines include interpolated LoRA weights fine-tuned separately on each objective using DDPO and AlignProp (Ramé et al., 2023; Clark et al., 2024; Prabhudesai et al., 2024), and a model directly fine-tuned on the weighted sum ($w = 0.5$). We use HPDv2 prompts for training and evaluation.

**Pareto-optimality without fine-tuning.** Figure 6a shows DAS achieving Pareto-optimal solutions without any fine-tuning or model interpolation, outperforming methods that require extensive training for each objective. While direct fine-tuning on weighted averages fails to improve the Pareto-front, DAS obtains optimal solutions for any reward combination by sampling from the reward-aligned target distribution. Figure 6b demonstrates this capability through superior prompt alignment and aesthetic quality across different reward weights.

## 4.3 ONLINE BLACK-BOX OPTIMIZATION

| Method | Target (↑) Aesthetic | Unseen Reward (↑) HPSv2 | Unseen Reward (↑) ImageReward | Diversity (↑) TCE | Diversity (↑) LPIPS |
|---|---|---|---|---|---|
| SEIKO-UCB | **6.88** | 0.25 | -0.31 | 38.5 | 0.51 |
| SEIKO-Bootstrap | 6.51 | 0.25 | -1.01 | 40.4 | 0.49 |
| **DAS-UCB (Ours)** | 6.77 | **0.28** | 1.25 | **41.1** | **0.66** |
| **DAS-Bootstrap (Ours)** | 6.73 | **0.28** | **1.27** | 40.6 | 0.65 |

Table 1: **Comparison of online optimization methods.** DAS achieves comparable target aesthetic scores while significantly outperforming in generalization to unseen rewards and output diversity.

**Online black-box optimization with diffusion models.** This approach optimizes an unknown function by receiving iterative feedback, especially useful when offline data is insufficient or objectives (e.g., human preferences) change over time. Minimizing feedback queries is key to reducing costs. SEIKO (Uehara et al., 2024b) is a feedback-efficient method using an uncertainty-aware optimistic surrogate model built through linear model (UCB) or ensembling (Bootstrap). While SEIKO guarantees theoretical regret bounds, this result relies on sampling from an aligned distribution using the surrogate model, similar to $p_{\text{tar}}$ in which they incorporate direct backpropagation to solve it. Instead, we adapt DAS to directly sample from this distribution (A.3).

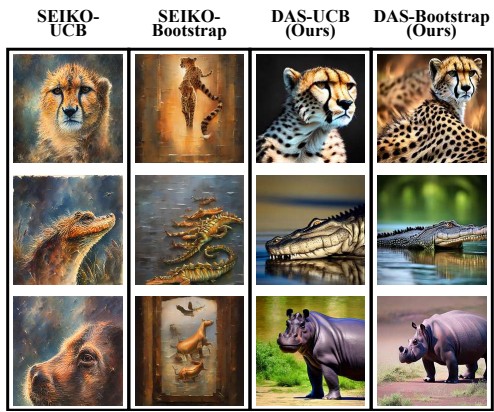

Figure 7: **Qualitative comparison of online methods.**

**Experiment setup.** We adopt aesthetic score as a black-box reward model, and limit to use only 1024 feedback queries for all methods. Experiment is conducted in a batch online setting through an iterative cycle: proposing samples (from fine-tuned model in SEIKO, or using DAS to directly sample from distribution aligned with surrogate model), recieving feedbacks from the black-box reward, and updating the surrogate model.

**Efficient exploration of diverse viable solutions.** Figure 7 highlights that DAS preserves pre-trained characteristics and generates diverse, high-quality images, while SEIKO, using AlignProp with KL, distorts animal features. Quantitatively, in Table 1, DAS matches SEIKO in optimizing aesthetic scores but significantly outperforms in unseen rewards and diversity, proving its ability to explore a broader solution space. This demonstrates the advantage of DAS over direct backpropagation for the online setting: high sample diversity enhances exploration, leading to a robust surrogate model and avoiding over-optimization. Furthermore, DAS bypasses fine-tuning the diffusion model every time the surrogate model is updated, enhancing adaptability through frequent updates.

**Non-differentiable rewards.** Reward maximization often involves non-differentiable or computationally expensive models. While DAS requires differentiable rewards for guidance, it can handle general rewards by posing the reward as black-box reward and learning a differentiable surrogate model with online feedback. We further demonstrate this using JPEG compressibility in G.3.

## 5 CONCLUSIONS

We introduce DAS, a test-time algorithm using Sequential Monte Carlo sampling to align diffusion models with rewards. DAS optimizes rewards while preserving generalization without fine-tuning. In single and multi-reward experiments, DAS achieves comparable or superior target rewards to fine-tuning methods while excelling in diversity and cross-reward generalization. The online optimization results demonstrate DAS's ability to efficiently explore diverse, high-quality solutions. These findings establish DAS as a versatile and efficient approach for aligning diffusion models applicable to a wide range of objectives and scenarios while significantly reducing the cost and complexity of the alignment process.

## REPRODUCIBILITY STATEMENT

We have made several efforts to ensure the reproducibility of our work. We provide complete proofs for all theoretical results in Appendix C, including formal statements and proofs for Propositions 2 and 3 in Appendix C.2.4 and C.2.5. Detailed pseudocode for our full DAS algorithm is included in Appendix A, with versions with adaptive resampling (Algorithm 1), adaptive tempering (Algorithm 3) and adaptation to online setting (Algorithm 5). Appendix D contains comprehensive implementation details for our method and baselines, including hyperparameter settings, training and sampling procedures. We will release our full codebase upon publication to enable others to replicate our results, including implementations of DAS. We use publicly available datasets and evaluation metrics, with details of experiment setup provided in Section 4. Appendix G contains additional experimental results to supplement those in the main paper. By providing these materials, we aim to enable other researchers to reproduce our results and build upon our work. We are committed to addressing any questions or requests for additional information to further support reproducibility efforts.

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

# A    PSEUDOCODES

## A.1    PSEUDOCODE FOR FULL ALGORITHM OF DAS

In practice, adaptive resampling (Chopin & Papaspiliopoulos, 2020; Murphy, 2023) is used instead of resampling every step with more sophisticated resampling schemes instead of multinomial resampling for variance reduction. We also used adaptive resampling with Srinivasan Sampling Process (SSP) resampling scheme (Gerber et al., 2019). The full algorithm of DAS is given as follows.

In practice, adaptive resampling (Chopin & Papaspiliopoulos, 2020; Murphy, 2023) is used instead of resampling at every step. This approach helpsmaintaining particle diversity where resampling every step may lead to particle degenarcy. Adaptive resampling uses the Effective Sample Size (ESS) as a criterion to determine when resampling is necessary. The ESS is defined as:

$$\text{ESS} = \left( \sum_{n=1}^{N} (W_t^n)^2 \right)^{-1} \tag{16}$$

where $W_t^n$ are the normalized particle weights. Resampling is performed only when the ESS falls below a predetermined threshold, indicating significant imbalance among particle weights. Note that weights in Equation 14 now become incremental weight.

Furthermore, more sophisticated resampling schemes are often employed instead of simple multinomial resampling to reduce variance. In our implementation of DAS, we use adaptive resampling with the Srinivasan Sampling Process (SSP) resampling scheme (Gerber et al., 2019).

The full algorithm of DAS incorporating these techniques is given in Algorithm 1 .

---

**Algorithm 1** Full Algorithm of DAS with Adaptive Resampling

1: **Input:** Number of time steps $T$, Number of particles $N$, Minimum ESS threshold $\text{ESS}_{\min}$, Resampling scheme $\text{RESAMPLE}(\cdot)$, Tempering scheme $0 = \lambda_T \leq \lambda_{T-1} \leq \cdots \leq \lambda_0 = 1$
2: **Output:** Particle approximations of $p_{tar}$, $\{X_0^n, W_0^n\}_{n=1}^N$
3: // Initialize particles at time T
4: **for** $n = 1$ to $N$ **do**
5:     $X_T^n \sim \mathcal{N}\left(0, \sigma_T^2 I\right)$                                    ▷ Sample from prior
6:     $w_T^n \leftarrow \exp\left(\frac{\lambda_T}{\alpha} \hat{r}(X_T^n)\right)$                           ▷ Initial weights
7: **end for**
8: $W_T^n \leftarrow w_T^n / \sum_{m=1}^N w_T^m$ for $n = 1, \ldots, N$                  ▷ Normalize weights
9: // Main loop: reverse time from T to 1
10: **for** $t = T$ to $2$ **do**
11:     // Adaptive resampling based on ESS
12:     $\text{ESS} \leftarrow \left( \sum_{n=1}^N (W_t^n)^2 \right)^{-1}$                            ▷ Calculate Effective Sample Size
13:     **if** $\text{ESS} < \text{ESS}_{\min}$ **then**
14:         $A_t^{1:N} \leftarrow \text{RESAMPLE}(W_t^{1:N})$                       ▷ Resample using SSP
15:         $\hat{w}_t^n \leftarrow 1$ for $n = 1, \ldots, N$                            ▷ Reset weights
16:     **else**
17:         $A_t^n \leftarrow n$ for $n = 1, \ldots, N$                              ▷ Keep original indices
18:         $\hat{w}_t^n \leftarrow w_t^n$ for $n = 1, \ldots, N$                       ▷ Keep original weights
19:     **end if**
20:     // Importance Sampling
21:     **for** $n = 1$ to $N$ **do**
22:         $X_{t-1}^n \sim m_{t-1}(x_{t-1} | X_t^{A_t^n})$                   ▷ Propose new particles, equation 13
23:         $w_{t-1}^n \leftarrow \hat{w}_t^n w_{t-1}(X_t^{A_t^n}, X_{t-1}^n)$                  ▷ Update weights, equation 14
24:     **end for**
25:     $W_{t-1}^n \leftarrow w_{t-1}^n / \sum_{m=1}^N w_{t-1}^m$ for $n = 1, \ldots, N$          ▷ Normalize weights
26: **end for**
27: **return** $\{X_0^n, W_0^n\}_{n=1}^N$

---

## A.2 Pseudocode for DAS with Adaptive Tempering

We also provide pseudocode of DAS with adaptive tempering used in ablation study (Section 4.1) for completeness.

---

**Algorithm 2** Full Algorithm of DAS with Adaptive Resampling and Adaptive Tempering

---

1: **Input:** Number of time steps $T$, Number of particles $N$, Target ESS $\text{ESS}_{\text{target}}$, Resampling scheme $\text{RESAMPLE}(\cdot)$
2: **Output:** Particle approximations of $p_{tar}$, $\{X_0^n, W_0^n\}_{n=1}^N$
3: // Initialize particles at time T
4: **for** $n = 1$ to $N$ **do**
5: $\quad X_T^n \sim \mathcal{N}\left(0, \sigma_T^2 I\right)$ $\hfill \triangleright$ Sample from prior
6: $\quad w_T^n \leftarrow 1$ $\hfill \triangleright$ Initial weights
7: **end for**
8: $W_T^n \leftarrow w_T^n / \sum_{m=1}^N w_T^m$ for $n = 1, \ldots, N$ $\hfill \triangleright$ Normalize weights
9: $\lambda_T \leftarrow 0$ $\hfill \triangleright$ Initial tempering parameter
10: // Main loop: reverse time from T to 1
11: **for** $t = T$ to 1 **do**
12: $\quad$ // Adaptive tempering
13: $\quad \delta \leftarrow \text{SOLVEFORDELTA}(\{X_t^n, W_t^n\}_{n=1}^N, \text{ESS}_{\text{target}}, \lambda_t)$ $\hfill \triangleright$ Solve for $\delta$
14: $\quad \lambda_{t-1} \leftarrow \min(\lambda_t + \delta, 1)$ $\hfill \triangleright$ Update tempering parameter
15: $\quad$ **for** $n = 1$ to $N$ **do**
16: $\quad\quad w_t^n \leftarrow w_t^n \cdot \exp\left(\frac{\lambda_{t-1} - \lambda_t}{\alpha} \hat{r}(X_t^n)\right)$ $\hfill \triangleright$ Update weights
17: $\quad$ **end for**
18: $\quad W_t^n \leftarrow w_t^n / \sum_{m=1}^N w_t^m$ for $n = 1, \ldots, N$ $\hfill \triangleright$ Normalize weights
19: $\quad$ // Adaptive resampling based on ESS
20: $\quad \text{ESS} \leftarrow \left(\sum_{n=1}^N (W_t^n)^2\right)^{-1}$ $\hfill \triangleright$ Calculate Effective Sample Size
21: $\quad$ **if** $\text{ESS} < \text{ESS}_{\text{target}}$ **then**
22: $\quad\quad A_t^{1:N} \leftarrow \text{RESAMPLE}(W_t^{1:N})$ $\hfill \triangleright$ Resample using SSP
23: $\quad\quad \hat{w}_t^n \leftarrow 1$ for $n = 1, \ldots, N$ $\hfill \triangleright$ Reset weights
24: $\quad$ **else**
25: $\quad\quad A_t^n \leftarrow n$ for $n = 1, \ldots, N$ $\hfill \triangleright$ Keep original indices
26: $\quad\quad \hat{w}_t^n \leftarrow w_t^n$ for $n = 1, \ldots, N$ $\hfill \triangleright$ Keep original weights
27: $\quad$ **end if**
28: $\quad$ // Importance Sampling
29: $\quad$ **if** $t > 1$ **then** $\hfill \triangleright$ Skip proposal step for the final iteration
30: $\quad\quad$ **for** $n = 1$ to $N$ **do**
31: $\quad\quad\quad X_{t-1}^n \sim m_{t-1}(x_{t-1} | X_t^{A_t^n})$ $\hfill \triangleright$ Propose new particles, equation 13
32: $\quad\quad\quad w_{t-1}^n \leftarrow \hat{w}_t^n w_{t-1}(X_t^{A_t^n}, X_{t-1}^n)$ $\triangleright$ Update weights using $\lambda_{t-1} = \lambda_t$, Eq. equation 14
33: $\quad\quad$ **end for**
34: $\quad\quad W_{t-1}^n \leftarrow w_{t-1}^n / \sum_{m=1}^N w_{t-1}^m$ for $n = 1, \ldots, N$ $\hfill \triangleright$ Normalize weights
35: $\quad$ **end if**
36: **end for**
37: **return** $\{X_0^n, W_0^n\}_{n=1}^N$

---

**Algorithm 3** SolveForDelta function with clamping

---

1: **function** $\text{SOLVEFORDELTA}(\{X_t^n, W_t^n\}_{n=1}^N, \text{ESS}_{\text{target}}, \lambda_t)$
2: $\quad$ Define $f(\delta) = \text{ESS}(\{W_t^n \cdot \exp(\frac{\delta}{\alpha} \hat{r}(X_t^n))\}_{n=1}^N) - \text{ESS}_{\text{target}}$
3: $\quad \delta_{\text{unclamped}} \leftarrow \text{NumericalRootFinding}(f)$ $\hfill \triangleright$ e.g., bisection method
4: $\quad \delta \leftarrow \max(0, \min(\delta_{\text{unclamped}}, 1 - \lambda_t))$ $\hfill \triangleright$ Clamp $\delta$ between 0 and 1-$\lambda_t$
5: $\quad$ **return** $\delta$
6: **end function**

---

A.3  PSEUDOCODE FOR ONLINE BLACK-BOX OPTIMIZATION

We first provide pseudocode of SEIKO (Uehara et al., 2024b) for completeness.

---

**Algorithm 4** SEIKO (OptimiStic finE-tuning of dIffusion with KL cOnstraint)

---

1: **Input:** Parameters $\alpha$, $\beta$, pre-trained diffusion model $f_{\text{pre}} : [0, T] \times X \to X$, initial distribution $\nu_{\text{pre}} : X \to \Delta(X)$
2: **Output:** Sequence of fine-tuned models $p^{(1)}, \ldots, p^{(K)}$
3: Initialize $f^{(0)} \leftarrow f_{\text{pre}}$, $\nu^{(0)} \leftarrow \nu_{\text{pre}}$
4: **for** $i = 1$ to $K$ **do**
5:      Generate new sample $x^{(i)} \sim \hat{p}^{(i-1)}(x)$ following reverse SDE:
6:          $dx_t = f^{(i-1)}(t, x_t)dt + \sigma(t)dw_t$, $x_0 \sim \nu^{(i-1)}$
7:      Get feedback $y^{(i)} = r(x^{(i)}) + \varepsilon$
8:      Update dataset: $D^{(i)} \leftarrow D^{(i-1)} \cup (x^{(i)}, y^{(i)})$
9:      Train surrogate model $\hat{r}^{(i)}(x)$ and uncertainty oracle $\hat{g}^{(i)}(x)$ using $D^{(i)}$
10:      Fine-tune diffusion model to fit the following distribution:
11:          $p^{(i)} \propto \exp\left(\frac{\hat{r}^{(i)}(\cdot) + \hat{g}^{(i)}(\cdot)}{\alpha + \beta}\right) \{\hat{p}^{(i-1)}(\cdot)\}^{\frac{\beta}{\alpha+\beta}} \{p_{\text{pre}}(\cdot)\}^{\frac{\alpha}{\alpha+\beta}}$
12:      and get $f^{(i)}, \nu^{(i)}$,
13: **end for**
14: **return** $p^{(1)}, \ldots, p^{(K)}$

---

For every update, SEIKO use direct backpropagation to fit the diffusion model to the target distribution $p^{(i)}$ by solving

$$p^{(i)} = \underset{p \in \Delta(\mathcal{X})}{\arg\max} \, \mathbb{E}_{x \sim p}[r(x)] - \alpha \text{KL}(p \parallel p^{(0)}) - \beta \text{KL}(p \parallel p^{(i-1)}) \tag{17}$$

similarly to Uehara et al. (2024a). However, we revealed that this method can easily fall into mode-seeking behavior in Section 3.2 and 4. Instead, we adapt DAS for the corresponding step to sample from

$$\tilde{p}^{(i)} \propto p_{\text{pre}}(\cdot) \exp\left(\frac{\hat{r}^{(i)}(\cdot) + \hat{g}^{(i)}(\cdot)}{\alpha}\right) \tag{18}$$

directly. Online algorithm employing DAS is given in Algorithm 5.

---

**Algorithm 5** Online Black-box Optimization using DAS

---

1: **Input:** Parameters $\alpha$, pre-trained diffusion model
2: **Output:** Samples from final target aligned to black box reward
3: Initialize $\tilde{p}^{(0)} = p_{\text{pre}}$
4: **for** $i = 1$ to $K$ **do**
5:      Generate new sample $x^{(i)} \sim \tilde{p}^{(i-1)}(x)$ using DAS (Algortihm 1):
6:      Get feedback $y^{(i)} = r(x^{(i)}) + \varepsilon$
7:      Update dataset: $D^{(i)} \leftarrow D^{(i-1)} \cup (x^{(i)}, y^{(i)})$
8:      Train surrogate model $\hat{r}^{(i)}(x)$ and uncertainty oracle $\hat{g}^{(i)}(x)$ using $D^{(i)}$
9: **end for**
10: **return** $p^{(1)}, \ldots, p^{(K)}$

---

For both aesthetic score experiments (Section 4.3 and JPEG compressibility experiments (SectionG.3), we used MLP on frozen CLIP embeddings (Radford et al., 2021) for surrogate model.

## B  INTRODUCTION TO SMC

In this section, we give introduction to SMC and particle filtering using Feyman-Kac formulation following Chopin & Papaspiliopoulos (2020). We additionaly introduce theoretical results in the SMC literature, which will be used in C.2.4 and C.2.5 for asymptotic analysis of DAS.

Please note that in this section, we follow similar time notation of general SMC methods which, unlike diffusion sampling, starts at time $0$ and ends at time $K$. We will come back to time notation for diffusion models in appendix C where we analyze DAS.

## B.1    FEYNMAN-KAC MODELS AND PARTICLE FILTERING

Feynman-Kac models extend a Markov process by incorporating potential functions to create a new sequence of probability measures via change of measure. Starting with a Markov process on state space $\mathcal{X}$, with initial distribution $\mathbb{M}_0(dx_0)$ and transition kernels $M_k(x_{k-1}, dx_k)$ for $k = 1 : K$, the joint distribution defined on $\mathcal{X}^K$ is:

$$\mathbb{M}_K(dx_{0:K}) = \mathbb{M}_0(dx_0) \prod_{k=1}^{K} M_k(x_{k-1}, dx_k) \tag{19}$$

The Feynman-Kac foramlization introduces potential functions $G_0(x_0)$ and $G_k(x_{k-1}, x_k)$ for $k \geq 1$ which are strictly positive. These define a new sequence of probability measures $\mathbb{Q}_k$ through change of measure:

$$\mathbb{Q}_k(dx_{0:k}) = \frac{1}{L_k} G_0(x_0) \left\{ \prod_{s=1}^{k} G_s(x_{s-1}, x_s) \right\} \mathbb{M}_k(dx_{0:k}) \tag{20}$$

refered as sequence of Feynman-Kac models. Here, $L_k$ is a normalizing constant:

$$L_k = \mathbb{E}_{\mathbb{M}_k} \left[ G_0(X_0) \prod_{s=1}^{k} G_s(X_{s-1}, X_s) \right] \tag{21}$$

Feynman-Kac model will also be used as a term that refer to the collection of kernels and potential functions $\{M_k, G_k\}$. The probability measures can be extended to an arbitrary future horizon $K \geq k$, allowing $Q_k$ to be defined on $\mathcal{X}^K$ for all $k$:

$$\mathbb{Q}_k(dx_{0:K}) = \frac{1}{L_k} G_0(x_0) \left\{ \prod_{s=1}^{k} G_s(x_{s-1}, x_s) \right\} \mathbb{M}_K(dx_{0:K}) \tag{22}$$

In this extended formulation, $\mathbb{Q}_k(dx_{0:K})$ is defined on the full time horizon $[0, K]$ for any $k \leq K$. $\mathbb{Q}_k(dx_{0:k})$ becomes the marginal distribution of the first $k + 1$ components of $\mathbb{Q}_k(dx_{0:K})$. For $s > k$, the potential functions $G_s$ are effectively set to 1, allowing the process to evolve according to the original Markov dynamics $M_s$ for the remaining time steps. Note that $\mathbb{Q}_k(dx_{0:k})$ is a marginal of $\mathbb{Q}_k(dx_{0:K})$, but not necessarily of $\mathbb{Q}_K(dx_{0:K})$ for $K > k$, a distinction crucial in working with Feynman-Kac models and related inference methods.

Sequential Monte Carlo (SMC), also known as particle filtering, is a generic algorithm that provides recursive approximations of a given state-space model. It relies on the Feynman-Kac model to provide the recursive structure for the probability distributions we wish to approximate, and uses importance sampling and resampling techniques to achieve these approximations. Algorithm 6 shows the sampling process of generic particle filtering algorithm.

---

**Algorithm 6** Generic Particle Filtering Algorithm

---

1: **Input:** Feynman-Kac model $\{M_k, G_k\}$, Number of particles $N$, Resampling scheme RESAMPLE($\cdot$), Number of time steps $K$
2: **Output:** Particle approximations $\{X_k^n, W_k^n\}_{n=1}^N$ for $k = 0, \ldots, K$
3: **for** $n = 1$ to $N$ **do**
4:     $X_0^n \sim M_0(dx_0)$
5:     $w_0^n \leftarrow G_0(X_0^n)$
6: **end for**
7: $W_0^n \leftarrow w_0^n / \sum_{m=1}^N w_0^m$ for $n = 1, \ldots, N$
8: **for** $k = 1$ to $K$ **do**
9:     $A_k^{1:N} \leftarrow$ RESAMPLE($W_{k-1}^{1:N}$)
10:     **for** $n = 1$ to $N$ **do**
11:         $X_k^n \sim M_k(X_{k-1}^{A_k^n}, dx_k)$
12:         $w_k^n \leftarrow G_k(X_{k-1}^{A_k^n}, X_k^n)$
13:     **end for**
14:     $W_k^n \leftarrow w_k^n / \sum_{m=1}^N w_k^m$ for $n = 1, \ldots, N$
15: **end for**
16: **return** $\{X_k^n, W_k^n\}_{n=1}^N$ for $k = 0, \ldots, K$

---

Ouput of the particle filtering algorithm can be used for sample approximations of the associated Feyman-Kac models by

$$\mathbb{Q}_{k-1}(dx_k) \approx \frac{1}{N} \sum_{n=1}^N \delta_{X_k^n} \tag{23}$$

$$\mathbb{Q}_k(dx_k) \approx \sum_{n=1}^N W_k^n \delta_{X_k^n} \tag{24}$$

$$\mathbb{E}_{\mathbb{Q}_{k-1}}[\varphi(X_k)] = \mathbb{Q}_{k-1} M_k(\varphi) \approx \frac{1}{N} \sum_{n=1}^N \varphi(X_k^n) \tag{25}$$

$$\mathbb{E}_{\mathbb{Q}_k}[\varphi(X_k)] = \mathbb{Q}_k(\varphi) \approx \sum_{n=1}^N W_k^n \varphi(X_k^n) \tag{26}$$

where $\varphi \in \mathcal{C}_b(\mathcal{X})$ is some test function and $\mathcal{C}_b(\mathcal{X})$ denotes the set of functions $\varphi : \mathcal{X} \to \mathbb{R}$ that are measurable and bounded. We give further asymptotic analysis of these approximations in the following sections when assuming multinomial resampling is used. For simplicity, we focus on the approximation 26.

Finally, we note that SMC samplers apply particle filtering where the associated Feynman-Kac model targets to approximate intermediate joint distributions in equation 7, but with different time notation.

### B.2  CONVERGENCE OF PARTICLE ESTIMATES

We state the following law of large number (LLN) type proposition for approximation 26 without proof. This proposition provides the asymptotic exactness of particle filtering algorithms and SMC samplers, including DAS, when the assumptions are met.

**Proposition 4** (Chopin & Papaspiliopoulos (2020), Proposition 11.4). *For algorithm 6 with multinomial resampling, if potential functions $G_k$'s of the associated Feynman-Kac model are all upper bounded, for $k \geq 0$ and $\varphi$ such that $\varphi \times G_k \in \mathcal{C}_b(\mathcal{X})$,*

$$\sum_{n=1}^N W_k^n \varphi(X_k^n) \xrightarrow{a.s.} \mathbb{Q}_k(\varphi). \tag{27}$$

*where $\xrightarrow{a.s.}$ denotes almost sure convergence.*

## B.3 CENTRAL LIMIT THEOREMS AND STABILITY OF ASYMPTOTIC VARIANCES

Even if approximations given by particle filtering algorithms and SMC samplers are asymptotically exact due to Proposition 4, the accuracy of approximation with finite number of particle depends on the rate of convergence. Typically, the asymptotic error is characterized by CLT type argument, where the error of estimation is distributed in Gaussian with scale $\mathcal{O}(N^{-\frac{1}{2}})$ and the asymptotic variance determines rate of convergence. We state formal version of the argument without proof.

**Proposition 5** (Chopin & Papaspiliopoulos (2020), Proposition 11.2). *Under the same settings and assumptions as Proposition 2, for $k \geq 0$ and $\varphi$ such that $\varphi \times G_k \in \mathcal{C}_b(\mathcal{X})$,*

$$\sqrt{N}\left(\sum_{n=1}^{N} W_k^n \varphi(X_k^n) - \mathbb{Q}_k(\varphi)\right) \Rightarrow \mathcal{N}(0, \mathcal{V}_k(\varphi)) \tag{28}$$

*where $\Rightarrow$ denotes convergence in distribution and the asymptotic variances $\mathcal{V}_k$'s are defined cumulatively as*

$$\mathcal{V}_k(\varphi) = \sum_{s=0}^{k} (\mathbb{Q}_{s-1} M_s) \left[(\bar{G}_s R_{s+1:k}(\varphi - \mathbb{Q}_k \varphi))^2\right]. \tag{29}$$

*Here, $\bar{G}_k = \frac{L_k}{L_{k-1}} G_k$, $R_k(\varphi) = M_k(\bar{G}_k \times \varphi)$ and $R_{s+1:k}(\varphi) = R_{s+1} \circ \cdots \circ R_k(\varphi)$.*

Due to the cumulative form of the asymptotic variance, it may easily blow up as the sampling errors accumulate. To prevent this, the Markov kernels should be strongly mixing, that is, future states of the Markov process should become increasingly independent of the initial state, making the effect of previous sampling errors vanish. We lay out the desired properties of the Markov kernels and potential functions, then state the stability of the asymptotic variance by providing an upper bound for the asymptotic variances that is uniform over time, based on the assumptions.

We first give a formal definition of strongly mixing Markov kernels.

**Definition 1** (Contraction coefficient, Strongly mixing Markov kernel). *The contraction coefficient of a Markov kernel $M_k$ is the quantity $\rho_M \in [0, 1]$ defined as*

$$\rho_M := \sup_{x_{k-1}, x'_{k-1}} \|M_k(x_{k-1}, dx_k) - M_k(x'_{k-1}, dx_k)\|_{TV}. \tag{30}$$

*where $\|\mathbb{P} - \mathbb{Q}\|_{TV} := \sup_{A \in \mathcal{B}(\mathcal{X})} |\mathbb{P}(A) - \mathbb{Q}(A)|$ is total variation distance between to probability measures $\mathbb{P}$ and $\mathbb{Q}$, and $\mathcal{B}(\mathcal{X})$ denotes Borel $\sigma$-algebra of state space $\mathcal{X}$. Furthermore, Markov kernel $M_k$ is said to be strongly mixing if $\rho_M \leq 1$.*

Next we lay out the assumptions for the associated Feynman-Kac model for the asymptotic variance to be stable.

**Assumption (1)** Markov kernels $M_k$ for $k = 1 : K$ admit a probability density $m_k$ such that

$$\frac{m_k(x_k|x_{k-1})}{m_k(x_k|x'_{k-1})} \leq c_M \tag{31}$$

for any $x_k, x_{k-1}, x'_{k-1} \in \mathcal{X}$, for some $c_M \geq 1$.

**Assumption (2)** Potential functions $G_k$'s are uniformly bounded for $k = 0 : K$ as

$$0 < c_l \leq G_k(x_{k-1}, x_k) \leq c_u \tag{32}$$

where $G_k(x_{k-1}, x_k)$ must be replaced by $G_0(x_0)$ for $k = 0$.

Given these assumptions, both $M_k$'s and Markov process defined by $\mathbb{Q}_k$'s become strongly mixing as below.

**Proposition 6** (Chopin & Papaspiliopoulos (2020), Proposition 11.9). *Under Assumptions (1) and (2), $M_k$ is strongly mixing with contraction coefficient contraction coefficient $\rho_M \leq 1 - c_M^{-1}$ Furthermore, the Markov process defined by $\mathbb{Q}_k$ is also strongly mixing with contraction coefficient $\rho_Q \leq 1 - 1/c_m^2 c_G$ where $c_G = c_u/c_l$*

Finally, using the assumptions and Proposition 3 one can prove the following proposition bounding the asymptotic variances uniformly in time.

**Proposition 7** (Chopin & Papaspiliopoulos (2020), Proposition 11.13). *Under Assumptions (1) and (2), for any $\varphi \in \mathcal{C}_b(\mathcal{X})$, asymptotic variance $\mathcal{V}_k(\varphi)$ define by 29 is bounded uniformly in time by*

$$\mathcal{V}_k(\varphi) \leq c_G^2 (\Delta\varphi)^2 \exp\left(\frac{2\rho_M c_G}{1 - \rho_Q}\right) \times \frac{1}{1 - \rho_Q^2} \tag{33}$$

$$\leq c_G^2 (\Delta\varphi)^2 \exp\left(\frac{2\left(1 - c_M^{-1}\right) c_G}{1 - \left(1 - (c_M^2 c_G)^{-1}\right)}\right) \times \frac{1}{1 - \left(1 - (c_M^2 c_G)^{-1}\right)^2} \tag{34}$$

*where $\Delta\varphi := \sup_{x,x' \in \mathcal{X}} |\varphi(x) - \varphi(x')|$ is the variation of $\varphi$.*

Note that the upper bound is increasing function of both $c_M$ and $c_G$. Intuitively, as these constants grow, the Markov kernels exhibit stronger mixing properties, which, in turn, accelerates the process of forgetting or diminishing the influence of past sampling errors. We will use this property in C.2.5 to prove that tempering can lower this uniform upper bound.

## C  PROOFS

### C.1  LOCALLY OPTIMAL PROPOSAL

In this section, we provide the analytic form of the unnormalized density of the locally optimal proposal, which is then approximated by Gaussian distribution for our proposal distribution. Before starting the proof, we note that one can alternatively get approximate samples from the locally optimal proposal using nested IS (Naesseth et al., 2019; Li et al., 2024) with additional use of samples. However, when the reward is differentiable, the Gaussian approximation works well without the need for additional samples, as we demonstrate in our experiments.

#### C.1.1  PROOF OF PROPOSITION 1

*proof.* We first show that $m_{t-1}^*(x_{t-1}|x_t)$ such that minimizes $\text{Var}(w_{t-1}(x_{t-1}, x_t)|x_t)$ satisfies $m_{t-1}^*(x_{t-1}|x_t) \propto \gamma_{t-1}(x_{t-1})L_t(x_t|x_{t-1})$. Since the minimization problem has constraint $\int dx_{t-1} m_{t-1}(x_{t-1}|x_t) = 1$, by introducing a Lagrange multiplier $\nu(x_t)$, the dual problem can be written as

$$\min_{m_{t-1}(x_{t-1}|x_t)} \left\{ \mathbb{E}_{m_{t-1}(x_{t-1}|x_t)}\left[w_{t-1}(x_{t-1}, x_t)^2\right] - \left(\mathbb{E}_{m_{t-1}(x_{t-1}|x_t)}\left[w_{t-1}(x_{t-1}, x_t)\right]\right)^2 \right.$$
$$\left. + \nu(x_t)\left(\int dx_{t-1} m_{t-1}(x_{t-1}|x_t) - 1\right)\right\}$$

Here, $w_{t-1}(x_{t-1}, x_t) = \frac{\bar{\pi}_{t-1}(x_{t-1:T})}{\bar{\pi}_t(x_{t:T})m_{t-1}(x_{t-1}|x_t)}$. Using calculation of variation and that only first and third term include $m_{t-1}$, $m_{t-1}^*$ should satisfy

$$0 = w_{t-1}^*(x_{t-1}, x_t)^2 - 2m_{t-1}^*(x_{t-1}|x_t)w_{t-1}^*(x_{t-1}, x_t)\frac{\bar{\pi}_{t-1}(x_{t-1:T})}{\bar{\pi}_t(x_{t:T})m_{t-1}^*(x_{t-1}|x_t)^2} + \nu(x_t)$$

$$= -w_{t-1}^*(x_{t-1}, x_t)^2 + \nu(x_t)$$

where $w_{t-1}^*(x_{t-1}, x_t) = \frac{\bar{\pi}_{t-1}(x_{t-1:T})}{\bar{\pi}_t(x_{t:T})m_{t-1}^*(x_{t-1}|x_t)}$. Since $\nu(x_t)$ is with constant respect to $x_{t-1}$,

$$m_{t-1}^*(x_{t-1}|x_t) \propto \frac{\bar{\pi}_{t-1}(x_{t-1:T})}{\bar{\pi}_t(x_{t:T})} \propto \gamma_{t-1}(x_{t-1})L_t(x_t|x_{t-1}). \tag{35}$$

Then using the definitions of intermediate targets and backward kernels used in DAS,

$$m_{t-1}^*(x_{t-1}|x_t) \propto p_\theta(x_{t-1}|x_t) \exp\left(\frac{\lambda_{t-1}}{\alpha}\hat{r}(x_{t-1})\right)$$

$$\propto \exp\left(-\frac{1}{2\sigma_t^2}\|x_{t-1} - \mu_\theta(x_t, t)\|^2 + \frac{\lambda_{t-1}}{\alpha}\hat{r}(x_{t-1})\right)$$

$\square$

## C.2 ASYMPTOTIC ANLYSIS OF DAS

### C.2.1 FEYNMAN-KAC MODEL FOR DAS

To give asymptotic analysis for DAS, we first clarify the Feynman-Kac model for DAS using the formulations from B. The Feynman-Kac model for DAS is given by simply substituting

$$M_0(dy_0) = p_T(x_T)(dx_T) = \mathcal{N}(0, \sigma_T^2 I)(dx_T) \tag{36}$$

$$M_k(y_{k-1}, dy_k) = m_{t-1}(x_{t-1}|x_t)(dx_{t-1}) = \mathcal{N}\left(\mu_\theta(x_t, t) + \sigma_t^2 \frac{\lambda_{t-1}}{\alpha} \nabla_{x_t} \hat{r}(x_t), \sigma_t^2 I\right)(dx_{t-1}) \tag{37}$$

$$G_0(y_0) = w_T(x_T) = \exp\left(\frac{\lambda_T}{\alpha} \hat{r}(x_T)\right) \tag{38}$$

$$G_k(y_{k-1}, y_k) = w_{t-1}(x_t, x_{t-1}) = \frac{p_\theta(x_{t-1}|x_t) \exp\left(\frac{\lambda_{t-1}}{\alpha} \hat{r}(x_{t-1})\right)}{m_{t-1}(x_{t-1}|x_t) \exp\left(\frac{\lambda_t}{\alpha} \hat{r}(x_t)\right)} \tag{39}$$

Then, the associated Feynman-Kac models are

$$\mathbb{Q}_{t-1}(dx_{t-1:T}) = \frac{1}{L_t} w_T(x_T) \left\{\prod_{s=t}^{T} w_{s-1}(x_s, x_{s-1})\right\} p_T(x_T) \prod_{s=t}^{T} m_{s-1}(x_{s-1}|x_s)(dx_{t-1:T})$$

$$= \bar{\pi}_T(x_T) \prod_{s=t}^{T} \frac{\bar{\pi}_{s-1}(x_{s-1:T})}{\bar{\pi}_t(x_{t:T}) m_{s-1}(x_{s-1}|x_s)} m_{s-1}(x_{s-1}|x_s)(dx_{t-1:T})$$

$$= \bar{\pi}_{t-1}(x_{t-1:T})(dx_{t-1:T})$$

where we used the alternative definition of $w_{t-1}$ in 8 and telescoping to simplify the terms. Thus indeed, Feynman-Kac models become the intermediate joint distributions defined through the intermediate targets and backward kernels. Especially, if we marginalize at $t = 0$, we get

$$\mathbb{Q}_0(dx_0) = \pi_0(x_0)(dx_0) = p_{tar}(x_0)(dx_0) \tag{40}$$

### C.2.2 ASSUMPTIONS FOR PROPOSITION 2 AND 3

Before we start the main proofs, we lay out the assumptions for the proofs.

**Assumption (a)** Reward function is bounded by $0 \leq r(\cdot) \leq R$

**Assumption (b)** Norm of gradient of $\hat{r}$ is uniformly bounded by $\|\nabla_{x_t} \hat{r}(x_t)\| \leq L$

**Assumption (c)** $\mathcal{X}_{t-1}$, defined as the union of support of $p_t$ and supports of $m_{t-1}(\cdot|x_t)$ for all $x_t$, is bounded and $d_{t-1} := \text{diam}(\mathcal{X}_{t-1}) = \sup\{d(x, y) : x, y \in \mathcal{X}_{t-1}\}$ for $t = 1 : T$.

We go over the viability of these assumptions. Assumption (a) can be satisfied lower and upper bounded rewards by adding a constant. Real-world rewards are indeed lower and upper bounded in most practical settings, including aesthetic score and PickScore used in our experiments. Even if not, we can simply clamp the reward to ensure the condition. Assumption (b) should be ensured for numerical stability of the algorithm. That is, if the gradient explode, generation using the guidance isn't possible. Since $r(\cdot)$ and $\hat{x}(\cdot)$ are function using neural networks, the assumption can be met unless $x_t$ gets out of support of the training distribution. Experimentally, this is commonly true unless using extremely small $\alpha$. Note that the assumption of uniform bound in time is only for simplicity and the bound may change in time. Assumption (c) is generally not true since $m_t$ is a Gaussian kernel and $p_t$ is also the marginal distribution over Gaussian noise added to clean data. However, it can 'effectively' be satisfied. We explain what this means in more detail. First, the data manifold $\mathcal{X}_0$ can be assumed to be bounded since most real-world data without corruption doesn't contain infinitely large or small component. Next, suppose we define the forward, reverse diffusion process and the Markov kernels using Gaussian distribution truncated at tail probability $\epsilon$ instead of standard Gaussian. Numerically, when $\epsilon$ is sufficiently small, the impact of the truncation becomes negligible, hence the diffusion process and the SMC sampler behaves similarly to the original Gaussian case. However, unlike the unbounded Gaussian noise, the bounded support of the truncated noise ensures compactness of $\mathcal{X}_t$ in Assumption (c). Thus the algorithm can be modified to satisfy Assumption (c) without any effect of the practical algorithm.

### C.2.3 LEMMAS

We first prove lemmas need to prove the propositions. We omit $t$ in $\mu_\theta(x_t, t)$ from now on for simplicity.

**Lemma 1.** *Under Assumptions (a) ~ (c),*

$$\frac{m_{t-1}(x_{t-1}|x_t)}{m_{t-1}(x_{t-1}|x_t')} \le \exp\left(\frac{d_{t-1}^2}{\sigma_t^2} + 3d_{t-1}\frac{\lambda_{t-1}}{\alpha}L + 2\left(\sigma_t\frac{\lambda_{t-1}}{\alpha}L\right)^2\right) \tag{41}$$

*for any $x_{t-1} \in \mathcal{X}_{t-1}$ and $x_t \in \mathcal{X}_t$*

*proof.*

$$\log \frac{m_{t-1}(x_{t-1}|x_t)}{m_{t-1}(x_{t-1}|x_t')}$$

$$= \frac{1}{2\sigma_t^2}\left(\|x_{t-1} - \mu_\theta(x_t) - \sigma_t^2\frac{\lambda_{t-1}}{\alpha}\nabla_{x_t}\hat{r}(x_t)\|^2 - \|x_{t-1} - \mu_\theta(x_t') - \sigma_t^2\frac{\lambda_{t-1}}{\alpha}\nabla_{x_t'}\hat{r}(x_t')\|^2\right)$$

$$= \frac{1}{\sigma_t^2}\left\langle x_{t-1} - \frac{1}{2}(\mu_\theta(x_t) + \mu_\theta(x_t')) - \frac{1}{2}\sigma_t^2\frac{\lambda_{t-1}}{\alpha}(\nabla_{x_t}\hat{r}(x_t) + \nabla_{x_t'}\hat{r}(x_t')),\right.$$

$$\left.(\mu_\theta(x_t) - \mu_\theta(x_t')) + \sigma_t^2\frac{\lambda_{t-1}}{\alpha}(\nabla_{x_t}\hat{r}(x_t) - \nabla_{x_t'}\hat{r}(x_t'))\right\rangle$$

Applying Cauchy-Schwarz inequality,

$$\le \frac{1}{\sigma_t^2}\left\|\left(x_{t-1} - \frac{1}{2}(\mu_\theta(x_t) + \mu_\theta(x_t')) - \frac{1}{2}\sigma_t^2\frac{\lambda_{t-1}}{\alpha}(\nabla_{x_t}\hat{r}(x_t) + \nabla_{x_t'}\hat{r}(x_t'))\right)\right\| \cdot$$

$$\left\|(\mu_\theta(x_t) - \mu_\theta(x_t')) + \sigma_t^2\frac{\lambda_{t-1}}{\alpha}(\nabla_{x_t}\hat{r}(x_t) - \nabla_{x_t'}\hat{r}(x_t'))\right\|$$

Using Assumption (b) to bound $\|\nabla_{x_t}\hat{r}(x_t)\|, \|\nabla_{x_t'}\hat{r}(x_t')\|$

and Assumption (c) to bound $\left\|x_{t-1} - \frac{1}{2}(\mu_\theta(x_t) + \mu_\theta(x_t'))\right\|, \|\mu_\theta(x_t) - \mu_\theta(x_t')\|$

since $x_{t-1}, \mu_\theta(x_t), \mu_\theta(x_t') \in \mathcal{X}_t$,

$$\le \frac{1}{\sigma_t^2}\left(d_{t-1} + \sigma_t^2\frac{\lambda_{t-1}}{\alpha}L\right) \cdot \left(d_{t-1} + 2\sigma_t^2\frac{\lambda_{t-1}}{\alpha}L\right)$$

$$= \frac{d_{t-1}^2}{\sigma_t^2} + 3d_{t-1}\frac{\lambda_{t-1}}{\alpha}L + 2\left(\sigma_t\frac{\lambda_{t-1}}{\alpha}L\right)^2$$

$\square$

Thus Assumption (1) in B.3 holds for Feynman-Kac model of DAS under Assumptions (a) ~ (c) where the uniform upper bound $c_M$ is given by

$$c_M = \sup_{t \in \{1,\dots,T\}}\left\{\exp\left(\frac{d_{t-1}^2}{\sigma_t^2} + 3d_{t-1}\frac{\lambda_{t-1}}{\alpha}L + 2\left(\sigma_t\frac{\lambda_{t-1}}{\alpha}L\right)^2\right)\right\}. \tag{42}$$

**Lemma 2.** *Under Assumptions (a) ~ (c),*

$$0 < \exp\left(-\left(d_{t-1} + \frac{1}{2}\sigma_t^2\frac{\lambda_{t-1}}{\alpha}L\right) \cdot \sigma_t^2\frac{\lambda_{t-1}}{\alpha}L - \frac{\lambda_t}{\alpha}R\right) \le w_{t-1}(x_t, x_{t-1}) \tag{43}$$

*and*

$$w_{t-1}(x_t, x_{t-1}) \le \exp\left(\left(d_{t-1} + \frac{1}{2}\sigma_t^2\frac{\lambda_{t-1}}{\alpha}L\right) \cdot \sigma_t^2\frac{\lambda_{t-1}}{\alpha}L + \frac{\lambda_{t-1}}{\alpha}R\right) \tag{44}$$

*proof.*

$$\log w_{t-1}(x_t, x_{t-1})$$

$$= \log \frac{p_\theta(x_{t-1}|x_t) \exp\left(\frac{\lambda_{t-1}}{\alpha}\hat{r}(x_{t-1})\right)}{m_{t-1}(x_{t-1}|x_t) \exp\left(\frac{\lambda_t}{\alpha}\hat{r}(x_t)\right)}$$

$$= \log p_\theta(x_{t-1}|x_t) - \log m_{t-1}(x_{t-1}|x_t) + \frac{\lambda_{t-1}}{\alpha}\hat{r}(x_{t-1}) - \frac{\lambda_t}{\alpha}\hat{r}(x_t)$$

By Assumption (a),

$$-\frac{\lambda_t}{\alpha}R \le \frac{\lambda_{t-1}}{\alpha}\hat{r}(x_{t-1}) - \frac{\lambda_t}{\alpha}\hat{r}(x_t) \le \frac{\lambda_{t-1}}{\alpha}R \tag{45}$$

Also,

$$\log p_\theta(x_{t-1}|x_t) - \log m_{t-1}(x_{t-1}|x_t)$$

$$= \frac{1}{2\sigma_t^2}\left(\|x_{t-1} - \mu_\theta(x_t)\|^2 - \left\|x_{t-1} - \mu_\theta(x_t) - \sigma_t^2\frac{\lambda_{t-1}}{\alpha}\nabla_{x_t}\hat{r}(x_t)\right\|^2\right)$$

$$= \frac{1}{\sigma_t^2}\left\langle x_{t-1} - \mu_\theta(x_t) - \frac{1}{2}\sigma_t^2\frac{\lambda_{t-1}}{\alpha}\nabla_{x_t}\hat{r}(x_t), \sigma_t^2\frac{\lambda_{t-1}}{\alpha}\nabla_{x_t}\hat{r}(x_t)\right\rangle$$

Applying Cauchy-Schwarz inequality,

$$\left|\left\langle x_{t-1} - \mu_\theta(x_t) - \frac{1}{2}\sigma_t^2\frac{\lambda_{t-1}}{\alpha}\nabla_{x_t}\hat{r}(x_t), \sigma_t^2\frac{\lambda_{t-1}}{\alpha}\nabla_{x_t}\hat{r}(x_t)\right\rangle\right|$$

$$\le \left\|x_{t-1} - \mu_\theta(x_t) - \frac{1}{2}\sigma_t^2\frac{\lambda_{t-1}}{\alpha}\nabla_{x_t}\hat{r}(x_t)\right\| \cdot \left\|\sigma_t^2\frac{\lambda_{t-1}}{\alpha}\nabla_{x_t}\hat{r}(x_t)\right\|$$

Using Assumption (b) to bound $\|\nabla_{x_t}\hat{r}(x_t)\|$

and Assumption (c) to bound $\|x_{t-1} - \mu_\theta(x_t)\|$ since $x_{t-1}, \mu_\theta(x_t) \in \mathcal{X}_t$

$$\le \left(d_{t-1} + \frac{1}{2}\sigma_t^2\frac{\lambda_{t-1}}{\alpha}L\right) \cdot \sigma_t^2\frac{\lambda_{t-1}}{\alpha}L$$

Combining the two bounds, we conclude the proof. $\qquad\square$

Thus Assumption (2) in B.3 also holds for Feynma-Kac model of DAS under Assumptions (a) $\sim$ (c). where $c_G = c_u/c_l$ is given by

$$c_G = \exp\left(\sup_{t\in\{1,\ldots,T\}}\left\{\left(d_{t-1} + \frac{1}{2}\sigma_t^2\frac{\lambda_{t-1}}{\alpha}L\right)\cdot\sigma_t^2\frac{\lambda_{t-1}}{\alpha}L + \frac{\lambda_{t-1}}{\alpha}R\right\} + \right.$$

$$\left. \sup_{t\in\{1,\ldots,T\}}\left\{\left(d_{t-1} + \frac{1}{2}\sigma_t^2\frac{\lambda_{t-1}}{\alpha}L\right)\cdot\sigma_t^2\frac{\lambda_{t-1}}{\alpha}L + \frac{\lambda_t}{\alpha}R\right\}\right) \tag{46}$$

### C.2.4 PROOF OF PROPOSITION 2

We state formal version of Proposition 2 and prove it.

**Proposition 8.** *For DAS with multinomial resampling, under Assumptions (a) $\sim$ (c), for $\varphi$ such that $\varphi \in \mathcal{C}_b(\mathcal{X}_0)$ and output of DAS $\{X_0^n, W_0^n\}_{n=1}^N$,*

$$\sum_{n=1}^N W_0^n\varphi(X_0^n) \xrightarrow{a.s.} p_{tar}(\varphi). \tag{47}$$

*where $p_{tar}$ is the final target distribution of DAS defined in 3.*

*proof.* By Lemma 2, each potential functions of the Feynma-Kac model are all upper bounded, and thus all conditions for Proposition 4 are met. Using Proposition 4 at $t = 0$ (i.e. $k = T$ respect to SMC for time notation), since $\mathbb{Q}_0(dx_0) = p_{tar}(x_0)(dx_0)$,

$$\sum_{n=1}^N W_0^n\varphi(X_0^n) \xrightarrow{a.s.} p_{tar}(\varphi). \tag{48}$$

$\square$

Setwise convergence of empirical measure can be derived as direct corollary by substituting $\varphi(X) = \mathbb{I}_A(X)$ for all $A \in \mathcal{B}(\mathcal{X}_0)$.

### C.2.5 PROOF OF PROPOSITION 3

Finally, we state formal version of Proposition 3 and prove it.

**Proposition 9.** *For DAS with multinomial resampling, under Assumptions (a) $\sim$ (c), for $\varphi$ such that $\varphi \in \mathcal{C}_b(\mathcal{X}_0)$ and output of DAS $\{X_0^n, W_0^n\}_{n=1}^N$,*

$$\sqrt{N} \left( \sum_{n=1}^N W_0^n \varphi(X_0^n) - p_{tar}(\varphi) \right) \Rightarrow \mathcal{N}(0, \mathcal{V}_0(\varphi)) \tag{49}$$

*where the asymptotic variance $\mathcal{V}_0(\varphi)$ is bounded by*

$$\mathcal{V}_0(\varphi) \leq c_G^2 (\Delta\varphi)^2 \exp \left( \frac{2 \left( 1 - c_M^{-1} \right) c_G}{1 - \left( 1 - (c_M^2 c_G)^{-1} \right)} \right) \times \frac{1}{1 - \left( 1 - (c_M^2 c_G)^{-1} \right)^2} \tag{50}$$

*using the definitions of $c_M$ and $c_G$ in 42 and 46. Furthermore, this upper bound when tempering is used, i.e. $\lambda_t$'s are not all 1 for $t = 0 : T$, is always smaller or equal to when tempering isn't used, i.e. $\lambda_t$'s are all 1 for $t = 0 : T$.*

*proof.* By Lemma 2, each potential functions of the Feynman-Kac model are all upper bounded, and thus all conditions for Proposition 5 are met. Using Proposition 5 at $t = 0$ (i.e. $k = T$ respect to SMC for time notation), since $\mathbb{Q}_0(dx_0) = p_{tar}(x_0)(dx_0)$,

$$\sqrt{N} \left( \sum_{n=1}^N W_0^n \varphi(X_0^n) - p_{tar}(\varphi) \right) \Rightarrow \mathcal{N}(0, \mathcal{V}_0(\varphi)) \tag{51}$$

Also, by Lemma 1 and 2 together, the Feynman-Kac model satisfies the Assumption (1) and (2) in B.3, thus using Proposition 7, we get

$$\mathcal{V}_0(\varphi) \leq c_G^2 (\Delta\varphi)^2 \exp \left( \frac{2 \left( 1 - c_M^{-1} \right) c_G}{1 - \left( 1 - (c_M^2 c_G)^{-1} \right)} \right) \times \frac{1}{1 - \left( 1 - (c_M^2 c_G)^{-1} \right)^2} \tag{52}$$

Looking at the definitions of $c_M$ and $c_G$ in 42 and 46, both values when tempering is used, i.e. $\lambda_t$'s are not all 1 for $t = 0 : T$, is always smaller or equal to when tempering isn't used, i.e. $\lambda_t$'s are all 1 for $t = 0 : T$ since the equations in the supremum are all increasing functions of $\lambda_t \geq 0$. Finally, since the upper bound is an increasing function of $c_M$ and $c_G$, we conclude that the upper bound when tempering is used is always smaller or equal to when tempering isn't used. $\square$

Again, similar result can be obtained for setwise convergence of empirical measure by substituting $\varphi(X) = \mathbb{I}_A(X)$ and $\Delta\varphi = 1$ for all $A \in \mathcal{B}(\mathcal{X}_0)$.

## D  IMPLEMENTATION DETAILS

In all experiments, we adapted Stable Diffusion (SD) v1.5 Rombach et al. (2022) for pre-trained model.

**Fine-tuning methods.** We used official PyTorch codebase of DDPO, AlignProp, TDPO with minimal change of hyperparameters from the settings in the original papers and codebases. We used 200 epoch and effective batch size of 256 using gradient accumulations if need for all methods. For AlignProp, even with KL regularization, severe reward collapse were mostly observed at the end of training, generating unrecognizable images. We used checkpoints before the collapse for comparisons. For AlignProp with KL regularization, we used the same coefficient of othe KL regularization terms. For DiffusionDPO, we used the official fine-tuned weights SD v1.5 using Pick-a-Pic dataset (Kirstain et al., 2023) released by the authors.

**Guidance methods.** We adapted the official PyTorch codebase of FreeDoM and MPGD to incorporate with diffusers library. For DPS, which wasn't adapted to latent diffusion, we used the same implementation of FreeDoM but withou time-travel startegy (Yu et al., 2023; Lugmayr et al., 2022). As in the official implementations, we scaled the guidance to match the scale of classifier-guidance and multiplied additional constants. These constants are 0.2 and 15 for FreeDoM and MPGD respectively, following the official implementation.

**DAS.** Across all experiment results except ablation studies, we used 100 diffusion time steps with $\gamma = 0.008$ for tempering. For single reward experiments, we used KL coefficient $\alpha = 0.01$ for aesthetic score task and $\alpha = 0.0001$ for PickScore task considering the scale of the rewards. For multi-objective experiments and online black-box optimization, we used $\alpha = 0.005$. We used 16 particles if not explicitly mentioned. Exceptionally, we used 4 particles during online black-box optimization for efficieny.

**DAS hyperparameter selection recipe.** To ehance easiness of adapting DAS, we propose a systematic approach for selecting hyperparameters based on empirical performance and convergence behavior. Firstly, tempering parameter $\gamma$ can be selected depending on the diffusion time steps $T$ such that $(1+\gamma)^T \approx 1$. While more particles are often better, 4 to 16 particles are sufficient to guarantee good performance as in the ablation study from Section 4.1. Especially, for rapid prototyping, we recommend 4 particles. KL coefficient $\alpha$ should be scaled to reward magnitude. Speicificaly, $\alpha$ in the range such that approximate guidance norm $\approx$ classifier guidance norm $* [1/5, 5]$ is appropriate. While $\alpha$ is the main tuning parameter, based on the above criteria, optimal values can be efficiently found through few sampling iterations since DAS requires no training. Typically, we fixed $T = 100$, $\gamma = 0.008$, $N = 4$ and used grid search for $\alpha \in \{10^{-1}, 10^{-2}, 10^{-3}, 10^{-4}\}$. After selecting the best $10^{-k}$, we again used grid search for $\alpha \in \{3 \times 10^{-(k+1)}, 5 \times 10^{-(k+1)}, 10^{-k}, 3 \times 10^{-k}, 5 \times 10^{-k}\}$.

# E   TEST-TIME SCALING

**Test-time Scaling.** While DAS offers a powerful test-time reward maximization scheme, even outperforming fine-tuning methods, it requires additional computation during inference. Similarly, recent literature on language models suggests that leveraging additional computation during inference can further improve the response's quality (Wei et al., 2022; Khanov et al., 2024; Wu et al., 2024; Snell et al., 2024). However, it's important to verify which method can most effectively utilize additional resources and achieve the highest performance as we scale test-time computation.

**Experiment Setup.** To demonstrate DAS is indeed an effective way to leverage computation during inference, we compare the test-time scaling of DAS with other possible test-time alignment methods for diffusion models. Baseline methods include Best-of-N (Touvron et al., 2023; Beirami et al., 2024; Ma et al., 2025) which runs $N$ (number of particles) independent generation process and selects the final sample with the highest reward. SMC uses a Sequential Monte Carlo sampler as DAS does, but without the approximate locally optimal proposal in Section 3.3.3 and tempering, using the original generation process as the proposal for each step. Including DAS, these methods are naturally scalable by increasing the number of particles used in the algorithm. We use the number of particles in $\{1, 2, 4, 8, 16, 32, 64, 128\}$ for each method and compare how target reward and unseen reward for cross-reward generalization scale as inference-time computation increase. We use PickScore for target reward and HPSv2 for unseen reward.

**Results.** Figure 8 shows that as the inference compute increases, DAS consistently outperforms both Best-of-N and SMC across both PickScore (target reward) and HPSv2 (unseen reward). Notably, DAS already achieves superior performance over other baselines even when using fewer particles and significantly less compute. This highlights DAS's efficiency in maximizing rewards compared to methods like Best-of-N, which shows limited improvements beyond a certain compute threshold, and SMC, which scales better than Best-of-N but still falls short of DAS.

These results emphasize the strength of the proposal and tempering scheme utilized by DAS, which enables it to effectively improve sample efficiency and thus leverage test-time computation without requiring extensive resources. By achieving better performance at lower computational cost and scaling seamlessly with increased resources, DAS proves to be both a compute-efficient and scalable method for inference-time alignment in diffusion models.

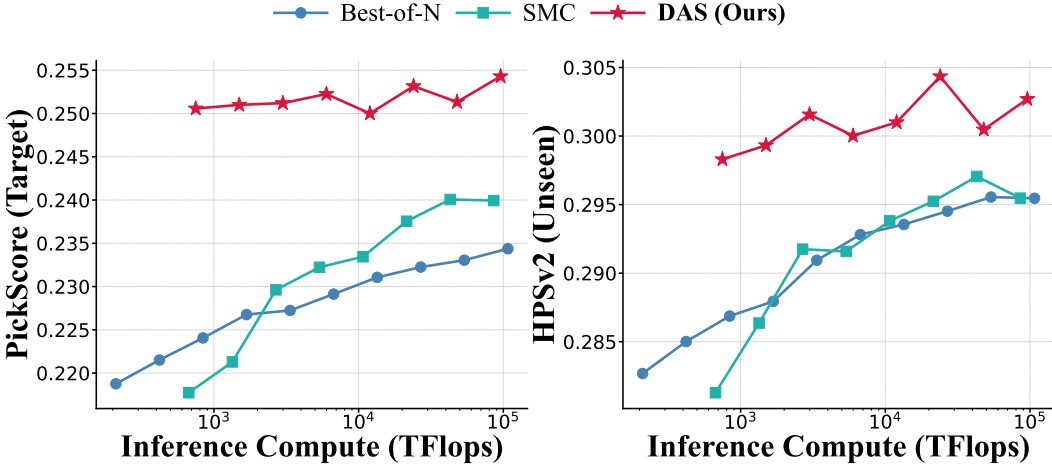

**Figure 8: DAS outperforms with fewer inference-time compute.** We compare DAS with other inference-time alignment algorithms while changing the inference-time compute, units in Teraflops. (left) Inference-time Compute vs. PickScore (target reward), (right) Inference-time Compute vs. HPSv2 (unseen reward)

## F    COMPARISON WITH DIFFUSION-BASED SAMPLERS

**Connection to Diffusion-based Samplers.** Starting from the objective of fine-tuning methods:

$$\underset{\theta}{\text{minimize}} D_{\text{KL}}(p_\theta \| p_{\text{tar}}), \tag{53}$$

RL (Fan et al., 2024) or direct backpropagation (Uehara et al., 2024a) optimize the variational lower bound of this objective given by the data processing inequality, $D_{\text{KL}}(p_\theta(x_{0:T}) \| p_{\text{tar}}(x_{0:T}))$, which is the KL divergence between joint distribution along the diffusion process (path measure for continuous processes). This objective can alternatively written as

$$D_{KL}(p_\theta(x_{0:T}) \| p_{\text{tar}}(x_{0:T})) = D_{KL}(p_\theta(x_{0:T}) \| p_{\text{ref}}(x_{0:T})) + \mathbb{E}_{x_0 \sim p_\theta(x_0)} \left[ \log \frac{p_{\text{tar}}(x_0)}{p_{\text{ref}}(x_0)} \right] \tag{54}$$

where $p_{\text{ref}}$ is defined by the same forward diffusion starting from a different reference distribution, since

$$p_{\text{tar}}(x_{0:T}) = p_{\text{tar}}(x_{0:T}|x_0)p_{\text{tar}}(x_0) = p_{\text{ref}}(x_{0:T}|x_0)p_{\text{tar}}(x_0) = p_{\text{ref}}(x_{0:T}) \frac{p_{\text{tar}}(x_0)}{p_{\text{ref}}(x_0)}. \tag{55}$$

In our problem setting for reward maximization, the reference diffusion is given by the pre-trained model by $p_{\text{ref}} = p_{\text{pre}}$ and $\log \frac{p_{\text{tar}}(x_0)}{p_{\text{ref}}(x_0)} = r(x_0)$, and thus the objective becomes reward with KL regularization term enforcing the diffusion process to stay close to the pre-trained diffusion process.

This formulation reveals the connection to diffusion-based samplers (Zhang & Chen, 2022; Vargas et al., 2023; Berner et al., 2024) which also use similar variational objective for training diffusion model to sample from a given unnormalized target density. While standard diffusion training can be interpreted as optimizing similar variational objective (Ho et al. (2020) for discrete time framework, Song et al. (2021b) for continuous time framework), they use conditional score matching using the samples from the target distribution. However, when sampling from unnormalized density, due to the lack of samples from the target, the methods use direct backpropagation for optimization. Similarly, reward alignment tasks also have no samples from the target distribution, thus previously proposed works are also based on direct backpropagation or RL. Compared to RL and direct backpropgation methods for reward maximization, diffusion-based samplers use different reference diffusions. Specifically, PIS (Zhang & Chen, 2022) use pinned Brownian motion running backwards in time, DDS (Vargas et al., 2023) use Variance Preserving (VP) SDE from Song et al. (2021c) (where DDPM is its discretization) as forward diffusion starting from a Normal distribution, and DIS (Berner et al., 2024) allows general reference diffusions. Note that given the same final target

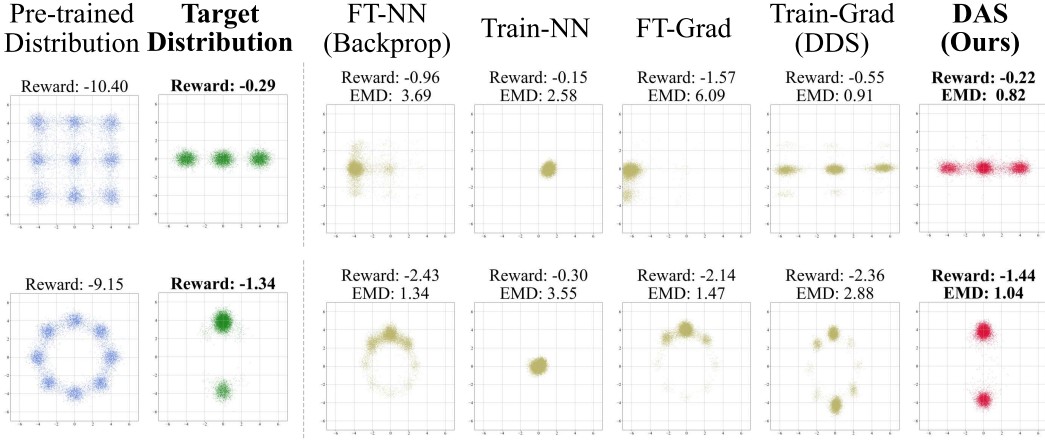

Figure 9: **Comparison with diffusion-based samplers.** Without either the Grad parameterization (Train-NN) or random initialization (FT-Grad), the methods fail to capture all modes in multimodal target distributions unlike DDS (Train-Grad), explaining the mode collapse of direct backpropagation (FT-NN). Furthermore, DAS still outperforms diffusion-based sampler for sampling from the target distribution

distribution, only the forward diffusion process matters for the training objective in equation 54. Thus when using DDPM sampling, i.e. VP SDE as forward diffusion, training of direct backpropagation methods, DDS, and DIS coincide.

**Key Differences.** In fact, the key distinction comes from the initialization of the models where diffusion-based samplers typically train from scratch, i.e. random initialization, while fine-tuning methods start from pre-trained models. Fine-tuning enables to incorporate prior knowledge from pre-trained model, offering a practical solution, especially for complex and high-dimensional data like images. However, it may make models more susceptible to mode collapse since it can only generate from the pre-trained distribution initially, as we demonstrate in our experiment below.

Additionally, diffusion-based samplers use model parameterization that incorporates the target score function. For example, PIS and DDS use $s_\theta(x_t, t) = \mathrm{NN}_1(x_t, t) + \mathrm{NN}_2(t) \cdot \nabla \log p_{\mathrm{tar}}(x)$, which we will refer as 'Grad parameterization'. Without Grad parameterization, diffusion-based samplers failed to fit multimodal target distribution, even for simple GMM (for example, Figure 2 in Zhang & Chen (2022) and Figure 13 in Berner et al. (2024)). However, we demonstrate that even if we incorporate Grad parameterization during fine-tuning, the model fails to capture all modes of the target, indicating the fundamental difference between fine-tuning and training from scratch.

**Experiment Setup.** We conducted additional experiments using GMM examples from Figure 1. To check our hypothesis and demystify the effect of each element, we conducted experiment using fine-tuning + NN parameterization (FT-NN, equivalent to direct backpropagation with KL regularization), fine-tuning + Grad parameterization (FT-Grad), training from scratch + NN parameterization (Train-NN), training from scratch + Grad parameterization (Train-Grad, equivalent to DDS). We excluded PIS from our comparison since DDS already outperforms PIS, where PIS uses pinned Brownian motion running backward in time as reference diffusion, which is proven to incur instability both theoretically (Appendix A.2 of Vargas et al. (2023)) and empirically (Appendix C.4 of Vargas et al. (2023)).

We used a two-layer architecture with 64 hidden units for all neural networks used in Figure 1 and 8 (for Grad parameterization, both $NN_1$ and $NN_2$), following Vargas et al. (2023). We used 100 diffusion time steps for DDPM sampling with linear beta schedule from 0.0001 to 0.02 as commonly used. For pre-training via conditional score matching, we used learning rate 0.001 with 1000 epochs. For DDS, we used learning rate $3\mathrm{e}^{-5}$ with 300 epochs. We used Adam optimizer for all training or fine-tuning. For other methods, we only changed epochs since each methods had different convergence rate when optimizing the objective. For example, the simple NN parameterization

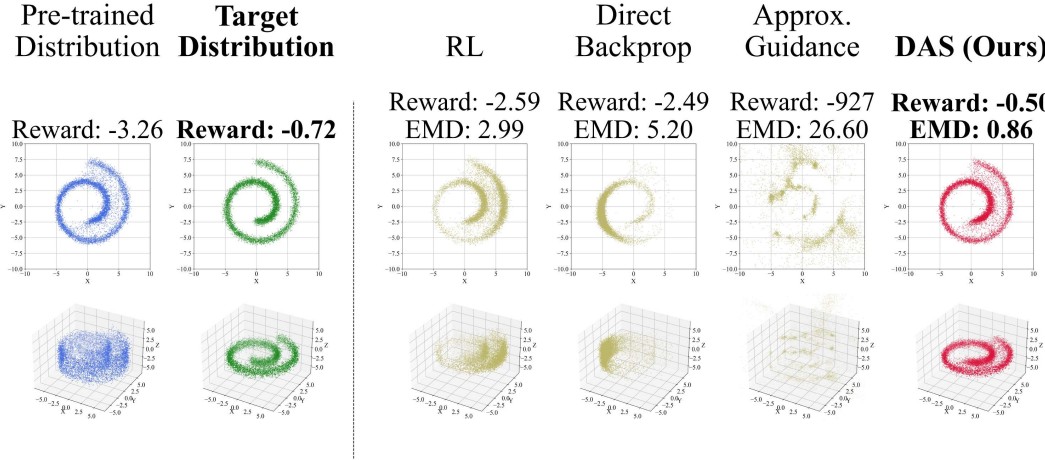

Figure 10: **SMC method excels in sampling from the target distribution compared to existing approaches.** Left of dashed line: Samples from pre-trained model trained on 3d Swiss roll, reward-aligned target distribution $p_{tar}$ using reward $r(X, Y, Z) = -X^2/100 - Y^2/100 - Z^2$. Right of dashed line: methods for sampling from $p_{tar}$ including previous methods (RL, direct backpropagation, approximate guidance) and ours using SMC. Top: Projection of samples to XY plane, reward, bottom: 3d plot of samples. EMD denotes sample estimation of Earth Mover's Distance, also known as Wasserstein distance between the sample distribution using each method and the target distribution. DAS outperforms existing approaches in capturing complex target distributions, as evidenced by lower EMD and the similarity with the target samples. Note that samples may exist outside the grid.

typically requires more iterations to converge compared to Grad parameterization, and training from scratch generally needs more iterations than fine-tuning.

**Empirical Validation.** The result presented in Figure 9 shows that the methods fail to capture all modes in multimodal target distributions without either the Grad parameterization from diffusion-based samplers or random initialization. Furthermore, DAS still outperforms DDS in EMD (earth mover's distance) between the target distribution, indicating that the samples from DAS are closer to the target distribution.

In conclusion, we claim that constraints in the reward alignment setting pose additional difficulty in training a diffusion model that can sample from unnormalized target distribution. DAS can overcome this problem by offering a training-free solution.

## G ADDITIONAL EXPERIMENT RESULTS

### G.1 3D SWISS ROLL

To further visualize the effectiveness of DAS for sampling from unnormlized target density using pre-trained diffusion model, we conducted additional experiment using 3d Swiss roll. As in Figure 10, DAS again demonstrates superior performance in sampling from the target distribution.

### G.2 DAS WITH SDXL

We conducted experiment using SDXL (Podell et al., 2024) as pre-trained base model to demonstrate the generality of our approach. We compare with the pre-trained SDXL (base + refiner), DPO-SDXL which fine-tuned SDXL using DiffusionDPO (Wallace et al., 2024), and our DAS with SDXL as base model. The results summarized in Figure 12 show that DAS's effectiveness generalizes beyond SD v1.5, achieving superior performance in both target optimization (PickScore) and cross-reward generalization (HPSv2, ImageReward) while maintaining competitive diversity metrics (TCE, LPIPS MPD), even with just 4 particles.

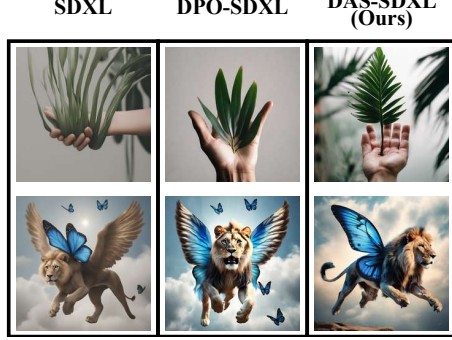

| | SDXL | DPO-SDXL (Ours) |
|---|---|---|

Figure 11: **Qualitative Comparison.**

| | SDXL | DPO | **DAS** |
|---|---|---|---|
| **PickScore*** | 0.23 | 0.23 | **0.25** |
| HPSv2 | 0.29 | 0.29 | **0.31** |
| ImageReward | 0.82 | 1.24 | **1.40** |
| TCE | 46.1 | **46.7** | 46.1 |
| LPIPS MPD | 0.57 | 0.61 | **0.61** |

*Target reward

Table 2: **Quantitative Comparison.**

Figure 12: **Experiment using SDXL.** Target reward: PickScore. Unseen prompts for qualitative comparison: "*A close up of a handpalm with leaves growing from it.*", "*A photo-realistic image of flying lion with blue butterfly wings*". Unseen prompts for quantitative comparison: HPDv2 evaluation prompts. Samples generated by DAS used only 4 particles.

| | SD1.5 | DAS-SD1.5 | SDXL | DAS-SDXL |
|---|---|---|---|---|
| **PickScore (Target)** | 0.220 | **0.256 (+0.036)** | 0.225 | **0.255 (+0.031)** |
| HPSv2 | 0.284 | **0.303 (+0.019)** | 0.285 | **0.306 (+0.021)** |
| ImageReward | 0.41 | **1.06 (+0.65)** | 0.82 | **1.40 (+0.58)** |
| TCE | 46.0 | **46.0** | 46.1 | **46.1** |
| LPIPS MPD | 0.662 | 0.647 | 0.568 | **0.608** |

Table 3: **Comparison of different backbones.** We compare DAS combined with SD1.5 and SDXL.

To check if the effectiveness of DAS depends on the pre-trained model's quality, we compared the performance of DAS when combined with SD1.5 and SDXL. As shown in Table 3, DAS improves the target reward score (PickScore) and cross reward scores (HPSv2 and ImageReward) while maintaining the diversity metrics (TCE and LPIPS MPD) regardless of the backbone's quality. Specifically, looking at the target reward (PickScore), DAS-SD1.5 achieves the most performance improvements, outpeforming the vanilla SDXL and performing nearly identical to DAS-SDXL. This indicates that the effectiveness of DAS does not depend on the quality of pre-trained models.

### G.3 NON-DIFFERENTIABLE REWARDS

As stated in Section 4.3, DAS can effectively optimize non-differentiable rewards by incorporating them as black-box rewards in our online black-box optimization framework. This allows us to handle non-differentiable objectives without modifying the core DAS algorithm. To further demonstrate this capability, we conducted experiments using JPEG compressibility - a strictly non-differentiable reward measured as the negative file size (in KB) after JPEG compression at quality factor 95. Our experimental setup included 4096 reward feedback queries, ImageNet animal prompts for evaluation, and comparison with DDPO (Black et al., 2023), which naturally handles non-differentiable rewards via RL. We combined DAS with UCB as described in Section 4.3.

The results in Figure 14 shows that DAS-UCB achieves the best compressibility score, outperforming both pre-trained model and DDPO. Also, it maintains CLIPScore and diversity metrics compared to the pre-trained model, mitigating over-optimization as intended. The qualitative results show how our method effectively minimizes background complexity while preserving key semantic features of the subjects.

In conclusion, while DAS is designed for differentiable rewards, our approach provides a practical and effective solution for non-differentiable objectives through online black-box optimization. The empirical results demonstrate that this approach outperforms methods specifically designed for non-

differentiable rewards while maintaining the key benefits of DAS such as diversity preservation and avoiding over-optimization.

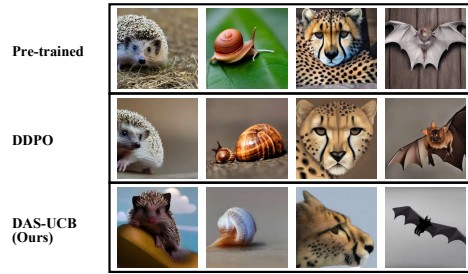

| | SD1.5 | DDPO | DAS |
|---|---|---|---|
| **Compressibility*** | -116 | -84 | **-71** |
| CLIPScore | **0.26** | 0.26 | 0.26 |
| TCE | 39.8 | 41.1 | **43.4** |
| LPIPS MPD | 0.66 | **0.62** | **0.62** |

Figure 13: **Qualitative Comparison.**

Table 4: **Quantitative Comparison.**

Figure 14: **JPEG Compressibility.** Online black-box optimization framework enable DAS to optimize non-differentiable rewards, outperforming DDPO which can naturally incorporate non-differentiable rewards using RL.

### G.4 AESTHETIC SCORE ADDITIONAL RESULTS

Figure 15 provides additional samples generated from each methods to target Aesthetic Score.

### G.5 PICKSCORE ADDITIONAL RESULTS

Figure 16 provides additional samples generated from each methods to target PickScore.

## H FUTURE WORKS

**Extended Analysis of Tempering.** While our theoretical and empirical analysis in Section 3.3 and 4.1 builds on both SMC literature for sampling efficiency and latent manifold deviation for understanding off-manifold behavior, recent work has uncovered additional interesting properties about non-uniform behavior (Wang et al., 2024; Zheng et al., 2025) and gradient contradictions (Hang et al., 2023; Go et al., 2024) in diffusion models. Our tempering approach already shows connections to these findings - the gradual adjustment of influence across timesteps aligns with the non-uniform importance discovered in recent works, and our careful tempering and resampling may naturally help mitigate gradient contradictions. Analyzing these connections more formally could provide additional theoretical insights complementing our existing analyses and further our understanding of why tempering proves particularly effective in the diffusion model context.

**Extensions to Other Modalities.** While we demonstrated DAS's effectiveness on image generation, our method could naturally extend to other modalities where diffusion models have shown promise. The core advantages of DAS - its training-free approach enabling quick adaptation to changing rewards, strong cross-reward generalization capabilities, and maintenance of sample diversity - make it particularly valuable for complex domains. For instance, in protein structure design, DAS could help optimize properties like binding affinity or stability while maintaining general protein functionality and exploring diverse structural variants. In audio generation, it could align generated speech or music with desired acoustic qualities while preserving broader sound characteristics and generalization across different quality metrics. For video generation, DAS could help optimize temporal consistency and visual quality across frames while maintaining robust performance across various video quality measures and diverse motion patterns. As these domains often involve complex, domain-specific rewards and constraints that may evolve during development, DAS's ability to generalize across rewards while preserving diversity–all without requiring any training–could significantly reduce development cycles while ensuring robust and versatile generation.

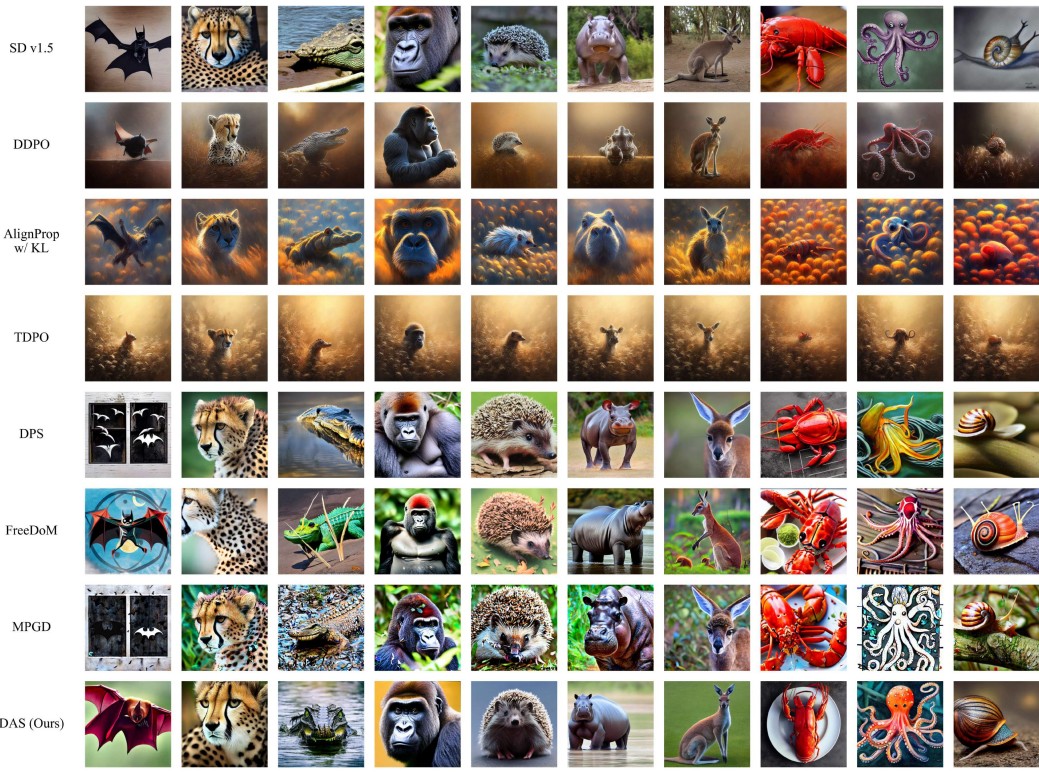

Figure 15: Images generated to target aesthetic score using prompts: "bat", "cheetah", "crocodile", "gorilla", "hedgehog", "hippopotamus", "kangaroo", "lobster", "octopus", "snail".

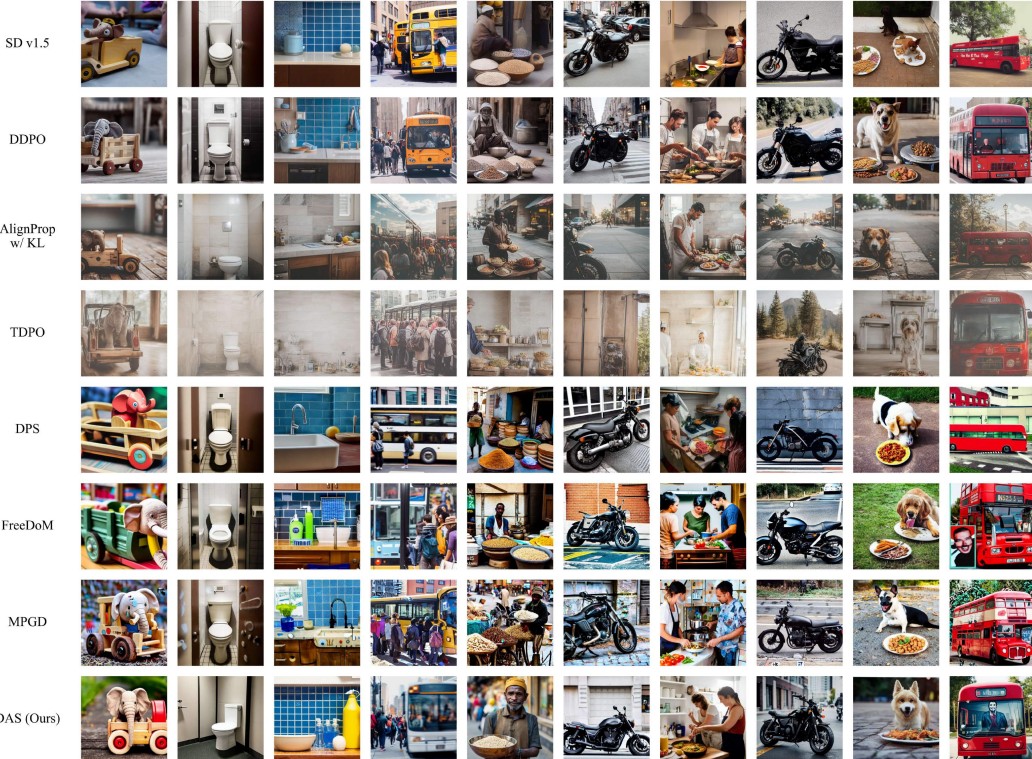

Figure 16: Images generated to target PickScore using prompts: "A toy elephant is sitting inside a wooden car toy.", "A white toilet in a generic public bathroom stall.", "An eye level counter-view shows blue tile, a faucet, dish scrubbers, bowls, a squirt bottle and similar kitchen items.", "People getting on a bus in the city.", "Street merchant with bowls of grains and other products.", "The black motorcycle is parked on the sidewalk.", "Three people are preparing a meal in a small kitchen.", "a black motorcycle is parked by the side of the road.", "a dog with a plate of food on the ground.", "there is a red bus that has a mans face on it."

