# OpenReview forum: "Test-time Alignment of Diffusion Models without Reward Over-optimization"
_ICLR.cc/2025/Conference — ICLR 2025 Spotlight_

### Official Review · Reviewer_31k3 · 2024-10-18

**Soundness:** 4
**Presentation:** 3
**Contribution:** 3
**Rating:** 8
**Confidence:** 3

**Summary:**

This paper propose a training-free sampling method based on Sequential Monte Carlo (SMC)  to align the diffusion models with specific objectives. Specially, Diffusion Alignment as Sampling (DAS) is designed to address the  limitations of the previous alignment approaches include fine-tuning and guidance methods. The author also provide theoretical analysis of DAS’s asymptotic properties and empirically validate DAS’s effectiveness across different tasks. Meanwhile, the authors conducte sufficient experiments to verify the validity of the DAS methodology.

**Strengths:**

The paper is clearly written and there is a good discussion of the work involved. Based on the fact that the existing fine-tuning methods lead to the reward overoptimization problem while the guidance methods lead to the under-optimization problem, the authors propose the DAS method to alleviate these deficiencies. In addition, the authors provide a theoretical analysis of the method and give the relevant code, making the work very solid. Figure 1 illustrates the shortcomings of the existing methods as well as the advantages of the proposed method, and the experimental results are visualized by using an example of a mixed Gaussian distribution.

**Weaknesses:**

+ The models underlying the experiments in this paper have some weaknesses, and the Stable Diffusion (SD) v1.5 model is somewhat outdated now. The Consistency model [1] and Flow model (SD3) [2]  are widely used nowadays, so I suggest the authors to conduct some experiments on the newer model so as to further illustrate the validity of the proposed method.

[1] Consistency models ICML-2024

[2] Scaling Rectified Flow Transformers for High-Resolution Image Synthesis ICML-2024

+ In addition to mixing Gaussian distributions, **Swiss rolls** are also commonly used to visualize whether a distribution has been learned or not, and due to their structural features, which can further reflect the model's ability to fit the distribution, the authors can give some visualizations that further illustrate the strengths of the proposed method.

**Questions:**

Please refer to Weaknesses. I will also refer to other reviewers' comments

---

> ### Author Response · Authors · 2024-11-21
>
> **W1.** `The models underlying the experiments in this paper have some weaknesses, and the Stable Diffusion (SD) v1.5 model is somewhat outdated now. The Consistency model [1] and Flow model (SD3) [2] are widely used nowadays, so I suggest the authors to conduct some experiments on the newer model so as to further illustrate the validity of the proposed method.
> `
>
> We thank the reviewer for your constructive comments. As per your suggestion, we extended (and are extending) our experiments with newer diffusion backbones. Firstly, we conducted additional experiments using SDXL, and the results are in the table below. Overall, **DAS still maintains its effectiveness with SDXL** in terms of both reward optimization (i.e., achieving the best target reward score; Pickscore) and mitigating over-optimization (i.e., maintaining or even improving non-target rewards and diversity metrics). This demonstrates that our method's core principles generalize well to more advanced architectures.
>
>
> |                    | SDXL (base + refiner) | DPO-SDXL | DAS-SDXl (Ours) |
> | ------------------ | --------------------- | -------- | --------------- |
> | PickScore (target) | 0.225                 | 0.232    | **0.255**       |
> | HPSv2              | 0.285                 | 0.294    | **0.306**       |
> | ImageReward        | 0.82                  | 1.24     | **1.40**        |
> | TCE                | 46.1                  | **46.7** | 46.1            |
> | LPIPS_MPD          | 0.567                 | 0.606    | **0.608**       |
>
> We are running experiments with newer architectures including the Consistency model and SD3, and will add this result as soon as the experiment is completed.
>
>
> **W2.** `In addition to mixing Gaussian distributions, Swiss rolls are also commonly used to visualize whether a distribution has been learned or not, and due to their structural features, which can further reflect the model's ability to fit the distribution, the authors can give some visualizations that further illustrate the strengths of the proposed method.`
>
> We appreciate the reviewer's valuable suggestion. Following this recommendation, we conducted additional experiments using the 3D Swiss rolls distribution. As shown in the table below, DAS again shows superior performance in sampling from the target distribution.
>
> |        | RL    | Direct Backprop | Approx. Guidance | DAS (Ours) |
> | ------ | ----- | --------------- | ---------------- | ---------- |
> | Reward | -2.59 | -2.49           | -927             | **-0.50**  |
> | EMD    | 2.99  | 5.20            | 26.69            | **0.86**   |
>
> Here, the Earth Mover's Distance (EMD) shows DAS achieves significantly better distribution matching compared to baselines, while maintaining better reward values. This further validates our method's ability to handle **complex and non-linear distributions**. We have **added the visualization results to the revised paper (Please see Figure 9 in Appendix F.1)**.

---

> ### Comment · Reviewer_31k3 · 2024-11-22
> **Thanks for the reply**
>
> Thanks to the author's careful reply, all my confusion are solved and I improve my score correspondingly. I also have a minor suggestion that some work finds that the training of diffusion models could be non-uniform [1,2] in different stages, that there may even be a gradient contradiction problem [3,4], and that this property may be useful for your work. Perhaps you could discuss them in a related work.
>
> [1] A Closer Look at Time Steps is Worthy of Triple Speed-Up for Diffusion Model Training arXiv-2024
>
> [2] Beta-tuned timestep diffusion model ECCV-2024
>
> [3] Efficient Diffusion Training via Min-SNR Weighting Strategy ICCV-2023
>
> [4] Addressing Negative Transfer in Diffusion Models NeurIPS-2023

---

> > ### Author Response · Authors · 2024-11-28
> > **Appreciation and Integration of Suggested References**
> >
> > We are glad that all of your confusion points have been resolved. Thank you again for your insightful comments.
> >
> > Furthermore, thank you for suggesting these highly relevant works. The non-uniform importance of different timesteps shown in [1,2] provides interesting potential connections to our tempering approach, which similarly adjusts influence gradually across timesteps. The gradient contradiction issues highlighted in [3,4] also suggest additional perspectives for understanding our SMC-based sampling method, as our tempering and resampling strategies may help mitigate such negative transfer effects. We have included analyzing these connections between DAS and diffusion model properties as future work (Appendix G).

---

> ### Author Response · Authors · 2024-12-04
> **Latent Consistency Model Results**
>
> We have completed additional experiments with the Latent Consistency Model (LCM) [a], a recent advancement in diffusion models that enables high-quality generation in very few steps. We used SimianLuo/LCM_Dreamshaper_v7 from huggingface as the pre-trained model.
>
> As shown in the table below, DAS successfully improves performance in both 4-step and 8-step inference settings, enhancing the target reward (PickScore, up to +0.011) while improving cross-reward generalization (HPSv2, ImageReward) and maintaining diversity metrics (TCE, LPIPS_MPD). These results further demonstrate that our method's effectiveness extends to modern architectures optimized for fast inference.
>
>
> |                    | LCM (4-steps) | DAS-LCM (4-steps, Ours) | LCM (8-steps) | DAS-LCM (8-steps, Ours) |
> | ------------------ | ------------- | ----------------------- | ------------- | ----------------------- |
> | PickScore (target) | 0.217         | **0.227 (+0.010)**       | 0.220         | **0.231 (+0.011)**      |
> | HPSv2              | 0.266         | **0.275 (+0.009)**      | 0.273         | **0.285 (+0.012)**      |
> | ImageReward        | 0.131         | **0.672 (+0.541)**      | 0.297         | **0.888 (+0.591)**      |
> | TCE                | 44.5          | 44.3                    | 45.1          | **45.5**                |
> | LPIPS_MPD          | 0.532         | **0.536**               | 0.530         | **0.535**               |
>
> [a] Luo, S., Tan, Y., Huang, L., Li, J., & Zhao, H. (2023). Latent consistency models: Synthesizing high-resolution images with few-step inference. arXiv preprint arXiv:2310.04378.

---

### Official Review · Reviewer_QHss · 2024-10-21

**Soundness:** 3
**Presentation:** 3
**Contribution:** 3
**Rating:** 8
**Confidence:** 3

**Summary:**

The paper introduces a novel, training-free sampling method aimed at generating samples from a target distribution, with a specific focus on applications in Reinforcement Learning from Human Feedback (RLHF). The authors compare their approach against existing fine-tuning baselines and guidance techniques.

**Strengths:**

- The introduction provides a clear overview of the problem.

- The proposed method appears promising and might be innovative (see Question 5.)

**Weaknesses:**

- The choice of finetuning-based RLHF baselines may not be appropriate (see Question 1).

- The paper is sometimes hard to follow due to the delayed definition of new notations. For instance, the symbol $\gamma$ is used on line 208 but is not defined until line 250.

- The evaluation metrics used in the paper (line 355 and onward) are not explained, making it difficult to assess their relevance and meaning.

**Questions:**

1. Baselines for Comparison: The authors correctly state that RLHF can be formulated as learning to sample from an unnormalized target distribution (Section 3.2). They show that current fine-tuning approaches in RLHF struggle to sample from multimodal target distributions, which highlights the limitations of these methods. However, this comparison may not be sufficient. There is significant research on using diffusion methods for sampling from multimodal distributions which would not fail at the examples presented in Figure 1 (e.g., [1], [2], [3] for continuous-time models, and [4] for discrete-time models). Including these approaches would provide a more convincing set of baselines. If this is not possible within the rebuttal phase's timeline, I believe it is necessary to at least mention this line of work in the paper.

2. Clarification on Calculations: The calculation presented on line 153 and the following lines is unclear. Can you provide a detailed derivation?

3. Explanation of Evaluation Metrics: The evaluation metrics mentioned in line 355 and onward lack a clear explanation. Currently, understanding them requires consulting multiple references. Could you include a brief explanation in the paper for clarity?

4. Inference Time Comparison: How does the inference time of your method compare with fine-tuning techniques? It seems plausible that fine-tuning methods might produce samples more quickly. Is this the case?

5. Novelty of the Method: Is the proposed method entirely new, or is it simply novel in the context of RLHF? How does it compare to other Sequential Monte Carlo methods?

I will initially give a score of 3, but I am willing to update my score if my questions are properly addressed.

# References
[1] Zhang, Qinsheng, and Yongxin Chen. "Path Integral Sampler: A Stochastic Control Approach For Sampling." International Conference on Learning Representations.

[2] Berner, Julius, Lorenz Richter, and Karen Ullrich. "An Optimal Control Perspective on Diffusion-Based Generative Modeling." Transactions on Machine Learning Research.

[3] Vargas, Francisco, Will Sussman Grathwohl, and Arnaud Doucet. "Denoising Diffusion Samplers." Eleventh International Conference on Learning Representations.

[4] Sanokowski, Sebastian, Sepp Hochreiter, and Sebastian Lehner. "A Diffusion Model Framework for Unsupervised Neural Combinatorial Optimization." Forty-First International Conference on Machine Learning.

[5] Dongjun Kim, Yeongmin Kim, Se Jung Kwon, Wanmo Kang, Il-Chul Moon Proceedings of the 40th International Conference on Machine Learning, PMLR 202:16567-16598, 2023.

---

> ### Author Response · Authors · 2024-11-21
> **Official Comment by Authors (1/3)**
>
> `The proposed method appears promising and might be innovative (see Question 5.)`
>
> Thank you for recognizing the potential of our method. We addressed your important question regarding novelty in our response to Question 5 by clarifying DAS's unique contributions over RLHF and sampling approahces. To further demonstrate its innovativeness, we highlight two additional key results included in our revised paper:
>
> 1. We have successfully applied DAS to **more recent pre-trained models**, e.g., SDXL, demonstrating its adaptability to state-of-the-art architectures (See Appendix F.2).
> 2. We have tested DAS with **non-differentiable rewards**, e.g., jpeg compressibility, showing that DAS is applicable to a wide range of reward functions (See Appendix F.3).
>
> Therefore, with the clarification and more results, we believe our approach is indeed innovative and has a broad applicability in many real-world applications.
>
>
> **W1.** `The choice of finetuning-based RLHF baselines may not be appropriate.`
> **Q1.** `Baselines for Comparison: The authors correctly state that RLHF can be formulated as learning to sample from an unnormalized target distribution (Section 3.2). They show that current fine-tuning approaches in RLHF struggle to sample from multimodal target distributions, which highlights the limitations of these methods. However, this comparison may not be sufficient. There is significant research on using diffusion methods for sampling from multimodal distributions which would not fail at the examples presented in Figure 1 (e.g., [1], [2], [3] for continuous-time models, and [4] for discrete-time models). Including these approaches would provide a more convincing set of baselines. If this is not possible within the rebuttal phase's timeline, I believe it is necessary to at least mention this line of work in the paper.`
>
> We sincerely thank the reviewer for this insightful comments regarding diffusion-based sampling methods. The comparison with these methods raises an important methodological question that deserves careful consideration.
>
> **Limtations of Diffusion-based Samplers.** Firstly, while [4] represents related work using diffusion models to sample from unnormalized target density, it is designed for discrete state spaces, making it unsuitable as a baseline for direct comparison. While [1, 2, 3] are indeed designed to sample from unnormalized target densities in continuous space, directly applying these works to our image tasks with $p_\text{tar} \propto p_\text{pre}\exp(r(x)/\alpha)$ presents significant challenges. These methods **train diffusion models from scratch**, demanding substantial computational resources and large datasets, and **backpropagate through the entire generation process** which becomes computationally prohibitive for large modern models, unless parameter-efficiently fine-tuning a pre-trained model using techniques like LoRA (Low Rank Adaptation) [a].
>
> **Adaptation to RLHF setting.** While directly applying diffusion-based samplers is impractical for general RLHF tasks for high-dimensional data like images, some technical aspects can be adapted to RLHF due to the **similarity of training objectives**. As we explain in detail in Appendix E in the revised paper, both fine-tuning methods and diffusion-based samplers optimze the same variational lower bound of the $D_{KL}(p_\theta \Vert p_{tar})$, where $p_\theta$ is the data distribution generated by the diffusion model and $p_{tar}$ is the target distribution. These methods use gradients of the objective as in the direct backpropagation fine-tuning method. While training from scratch is impractical for general settings as explained, the parameterization proposed in these methods can be utilized. Specifically, all [1, 2, 3] use **parameterization that incorporate score function of the target distribution**: $s_\theta(x_t,t) = \text{NN}_1(x_t,t) + \text{NN}_2\cdot\nabla \log p_\text{tar}(x)$, which will refer as 'Grad parameterization'. This parameterization is key to the success of diffusion-based samplers. Figure 2 in [1] and Figure 13 in [2] shows that without such paramterization, the method fails to capture all modes of a multimodal target distribution. Thus, we conduct additional experiments to validate whether such parameterization can help enhance fine-tuning methods using GMM examples from Figure 1.
>
> **Experiments.** To orthogonally analyze the effect of fine-tuning and Grad parameterization, we use fine-tuning + NN parameterization (FT-NN, equivalent to Direct Backprop w/ KL regularization),  fine-tuning + Grad parameterization (FT-Grad),  training from scratch + NN parameterization (Train-NN), training from scratch + Grad parameterization (Train-Grad, equivalent to diffusion-based sampler with the pre-trained model as reference diffusion; please refer to Appendix E for definition of reference diffusion) in addition to DAS for comparison. The results are summarized in the table below:
>
> (Continued in the next comment)

---

> ### Author Response · Authors · 2024-11-21
> **Official Comment by Authors (2/3)**
>
> |                        | FT-NN (Direct Backprop) | Train-NN | FT-Grad | Train-Grad (Sampler) | DAS (Ours) |
> | ---------------------- | ----------------------- | -------- | ------- | -------------------- | ---------- |
> | Reward (Ex1)           | -0.96                   | -0.15    | -1.57   | -0.55                | **-0.22**  |
> | EMD (Ex1)              | 3.69                    | 2.58     | 6.09    | 0.91                 | **0.82**   |
> | Reward (Ex2)           | -2.43                   | -0.30    | -2.14   | -2.36                | **-1.44**  |
> | EMD (Ex2)              | 1.34                    | 3.55     | 1.47    | 2.88                 | **1.04**   |
> | Captures Multimodality | X                        | X         | X        | **O**                     | **O**           |
>
> The results show that even with Grad parameterization, the fine-tuning method fails to capture all modes.  Furthermore, DAS still outperforms diffusion-based samplers for sampling from the target distribution. The experiment highlights the difficulty of RLHF problem for diffusion models: **While fine-tuning a pre-trained model is inevitable when learning reward-aligned target distribution, it increases susceptibility to mode collapse.** This also hints how **DAS avoid mode collapse and over-optimization through training-free solution**.
>
> We have also incorporated references [1~4] as related works in Section 2.1, discussing their relationship to fine-tuning methods.
>
> We are grateful to the reviewer for the valuable references, which helped improve our literature review and strengthen our claims about existing methods.
>
> ---
> [a] Hu, E. J., Wallis, P., Allen-Zhu, Z., Li, Y., Wang, S., Wang, L., & Chen, W. LoRA: Low-Rank Adaptation of Large Language Models. In International Conference on Learning Representations.
>
> **W2.** `The paper is sometimes hard to follow due to the delayed definition of new notations. For instance, the symbol` $\gamma$ `is used on line 208 but is not defined until line 250.`
>
> We sincerely apologize for any confusion caused by delayed definitions. We appreciate you pointing out this issue. Following your feedback, we have added clear definitions for all symbols at their first appearance including $\gamma$, to help readers better follow the paper (line 183, 209, 233, 253). We hope these changes improve the paper's readability.
>
>  **W3.** `The evaluation metrics used in the paper (line 355 and onward) are not explained, making it difficult to assess their relevance and meaning.`
>  **Q3.** `Explanation of Evaluation Metrics: The evaluation metrics mentioned in line 355 and onward lack a clear explanation. Currently, understanding them requires consulting multiple references. Could you include a brief explanation in the paper for clarity?`
>
> Thank you for suggesting the need for clearer explanations of the evaluation metrics. In the revised paper, we have added explanations for each metric and clarified how they help identify over-optimization (Section 4.1.1, lines 354~359). Specifically, we explain that low scores in cross-reward generalization or diversity suggest over-optimization, and describe how each metric evaluates different aspects: HPSv2 and ImageReward for image quality and text-image alignment, TCE for semantic diversity, and LPIPS for perceptual differences between samples. We hope these additions make the evaluation framework clearer and more accessible.
>
>  **Q2.** `Clarification on Calculations: The calculation presented on line 153 and the following lines is unclear. Can you provide a detailed derivation?`
>
> We appreciate your careful review which helps make our paper more rigorous and accessible. Let us provide a detailed derivation of the calculations.
>
> We introduce a binary variable $\mathcal{O} \in \{0, 1\}$ that indicates whether a sample $x$ is "optimal" ($\mathcal{O}(x) = 1$) or "not optimal" ($\mathcal{O}(x) = 0$). That is, $\mathcal{O}$ is a Bernoulli random variable conditioned on $x$. Next, we define the probability of optimality for any sample $x$ as: $p(\mathcal{O} = 1|x) = \frac{1}{C}  \exp(r(x)/α)$ (note that the normalization constant $C$ doesn't matter in our discussion). Then, samples with high reward are interpreted as more likely "optimal", with $\alpha$ controlling the strictness of optimality.
> Conversly, the posterior $p(x|\mathcal{O}=1)$ conditioned on optimality characterizes the distribution of samples that achieve high rewards. Using Bayes' rule with prior $p=p_\text{pre}$:
> $$p(x|O = 1) = \frac{p(x)p(O = 1|x)}{p(O = 1)} \propto p_\text{pre}(x)\exp\left(\frac{r(x)}{\alpha}\right) \propto p_\text{tar}(x)$$, we get back our original target distribution.
> Therefore, the two perspectives: reward maximization with KL regularization vs. sampling from the posterior $p(x|\mathcal{O}=1)$, are equivalent.
>
> We added the detailed explanation in the revised paper (line 152~157) to make it more accessible to readers. Thank you for helping us identify this gap in our exposition.

---

> ### Author Response · Authors · 2024-11-21
> **Official Comment by Authors (3/3)**
>
> **Q4.** `Inference Time Comparison: How does the inference time of your method compare with fine-tuning techniques? It seems plausible that fine-tuning methods might produce samples more quickly. Is this the case?`
>
> Thank you for raising this important point about computational efficiency. We acknowledge that our method is computationally more intensive at inference time compared to fine-tuned models, due to the need for multiple particles (though as shown in Figure 5, just 4 particles are sufficient for strong performance) and additional computation of guidance terms for each sampling step. Comparison with baseline methods using 100 DDIM steps on a single A100 GPU with SD1.5 as pre-trained model, aesthetic score as reward, and "cheetah" for the prompt is organized in the table below:
>
> |                | SD1.5 | Fine-tuned | FreeDoM [b] | DAS (Ours) |
> | -------------- | ----- | ---------- | ------- | ---------- |
> | GPU Time (sec) | 4     | 4          | 25      | 66         |
>
>
> However, we believe that the benefits DAS offers in certain applications outweigh this overhead.
>
> - **Adaptability to changing reward function**. In scenario where the reward function may change, fine-tuning the model each time is computationally expensive. Specifically, in online settings with evolving reward, DAS is particularly useful as it bypasses the need for iterative fine-tuning. Our results also highlight the effectiveness of DAS in multi-reward scenarios (Section 4.2). For prototyping combinations of multiple reward functions, DAS offers an efficient solution without training overhead.
> - **Diversity**. In applications like creative content generation and drug discovery, the diversity of generated samples is crucial for fostering innovation and uncovering novel solutions. We have validated that while fine-tuning approaches often lead to limited diversity due to over-optimization, DAS maintains high levels of diversity in the generated samples.
>
> ---
>
> [b] Yu, J., Wang, Y., Zhao, C., Ghanem, B., & Zhang, J. (2023). Freedom: Training-free energy-guided conditional diffusion model. In Proceedings of the IEEE/CVF International Conference on Computer Vision
>
> **Q5.** `Novelty of the Method: Is the proposed method entirely new, or is it simply novel in the context of RLHF? How does it compare to other Sequential Monte Carlo methods?`
>
>
> Thank you very much to help us improve our paper. Our proposed approach is not simply applying SMC in the context of RLHF, but entirely novel in three ways:
> - **1) Introduction of tempering technique.** We introduce tempering into the SMC framework, which is a *key* to solve the reward-aligned sampling in the context of RLHF. As shown in Figure 5 in our abalation study, SMC method without tempering fails to sample from the target distribution, leading to reward under-optimization even with 32 particles. In contrast, DAS with tempering successfully samples from the target distribution with only 4-8 particles, achieving both high rewards and good generalization.
> - **2) Theoretical justification.** We further provide theoretical analysis that prove tempering is essential for *exact sampling* from the target distribution. Specifically, we
> provide *theoretical guarantees* showing that tempering reduces asymptotic variance (Proposition 3) by mitigating weight degeneracy and off-manifold guidance.
> - **3) Practical efficiency** Unlike previous works that apply SMC to diffusion sampling that require *dozens to hundreds of particles* [c, d, e], our method achieves strong, saturated results with *just 4-8 particles* as our ablation study shows. This was possible through our theoretically-motivated design choices in tempering and proposal distributions. Such efficiency is especially important when scaling to large pre-trained models like SDXL (we extended our framework with SDXL which is in Appendix F.2 during the rebuttal period).
>
> We believe these innovations, particularly our tempering framework for diffusion sampling, enable effective sampling from reward-aligned target distributions, as demonstrated by our strong empirical results across single-reward, multi-reward, and online optimization tasks. We thank the reviewer for encouraging us to clarify these distinctions.
>
>
> ---
>
> [c] Trippe, B. L., Yim, J., Tischer, D., Baker, D., Broderick, T., Barzilay, R., & Jaakkola, T. S. Diffusion Probabilistic Modeling of Protein Backbones in 3D for the motif-scaffolding problem. In The Eleventh International Conference on Learning Representations.
>
> [d] Wu, L., Trippe, B., Naesseth, C., Blei, D., & Cunningham, J. P. (2024). Practical and asymptotically exact conditional sampling in diffusion models. Advances in Neural Information Processing Systems, 36.
>
> [e] Dou, Z., & Song, Y. (2024). Diffusion posterior sampling for linear inverse problem solving: A filtering perspective. In The Twelfth International Conference on Learning Representations.

---

> > ### Comment · Reviewer_QHss · 2024-11-22
> > **Raised score and further questions**
> >
> > I thank the authors for the detailed response.
> > My questions are adequately addressed and I will therefore raise my score from 3 to 6.
> >
> > However, I think the newly added Experiment in App.E misses a lot of experimental details and I would appreciate it if these will be added.
> > I would appreciate it if the authors would add the following details:
> > - which diffusion sampler is exactly used and what hyperparameters are used for training of the diffusion sampler
> >
> > I also do not understand why the diffusion sampler (Train-Grad if I understood correctly) captures a mixture of the target and pre-trained distribution, although it was trained from scratch (i.e. has no information of the pre-trained distribution). Or did I misunderstand something in the description of this experiment?
> >
> > If these questions are addressed I will consider raising my score to 8.

---

> > > ### Author Response · Authors · 2024-11-24
> > > **Experiment details in App.E**
> > >
> > > Thank you for your thoughtful feedback and for taking the time to review our revised submission. We greatly appreciate your detailed comments and are pleased that our previous response helped address your initial concerns. We'll now address your remaining questions about the experimental details and the diffusion sampler behavior.
> > >
> > > `However, I think the newly added Experiment in App.E misses a lot of experimental details and I would appreciate it if these will be added. I would appreciate it if the authors would add the following details: which diffusion sampler is exactly used and what hyperparameters are used for training of the diffusion sampler`
> > >
> > > We apologize for any lack of clarity about the baseline diffusion sampler. To be more specific, **the baseline we used is equivalent to DDS** [3]. For clarity, we've updated the name to "Train-Grad (DDS)" in Appendix E of the revised paper.
> > >
> > > Next we list the experiment details and hyperparameters of the methods. For all experiments in Figures 1 and 8, we used the following settings:
> > >
> > > **1. Neural Network Architecture:**
> > > - Two-layer architecture with 64 hidden units for all networks (including both $NN_1$ and $NN_2$ for Grad parameterization), following [3]
> > >
> > > **2. Diffusion Sampling settings:**
> > > - 100 diffusion time steps with DDPM sampling
> > > - Linear beta schedule from 0.0001 to 0.02
> > >
> > > **3. Training settings:**
> > > - Adam optimizer throughout
> > > - Pre-training via conditional score matching: learning rate 0.001, 1000 epochs
> > > - DDS: learning rate 3e-5, 300 epochs
> > > - FT-Grad: learning rate 3e-5, 200 epochs
> > > - Train-NN: learning rate 3e-5, 2000 epochs
> > > - FT-NN (Direct Backprop): learning rate 3e-5, 500 epochs
> > >
> > > While all methods except pre-training optimize the same training objective, we adjusted the number of epochs based on their different convergence behaviors. For example, the simple NN parameterization typically requires more iterations to converge compared to Grad parameterization, and training from scratch generally needs more iterations than fine-tuning.
> > >
> > > We added this in 'Experiment Setting' in Appendix E (lines 1496~1504) of the newly revised paper.
> > >
> > > We would also like to explain our choice of DDS as the baseline and clarify its equivalence to Train-Grad:
> > >
> > > 1. Regarding our baseline selection:
> > >
> > > - DIS with Variance Preserving (VP) SDE forward diffusion process (where DDPM is its discretization) is equivalent to DDS, as proven in Appendix A.10.1 of [2]
> > > - Since our KL-regularized baselines use DDPM sampling, and DIS [2] also used DDPM sampling in their experiments, we chose to include DDS as our baseline
> > > - We excluded PIS from our comparison since DDS already outperforms PIS, where PIS uses pinned Brownian motion running backwards in time as reference diffusion, which is proven to incur instability both theoretically (Appendix A.2 of [3]) and empirically (Appendix C.4 of [3])
> > >
> > >
> > > 2. The equivalence between Train-Grad and DDS comes from two key points:
> > >
> > > - Both RLHF with KL regularization and DDS optimize the same training objective:
> > > $$D_{KL}(p_\theta(x_{0:T})||p_\text{tar}(x_{0:T}))$$
> > > - Both use identical components:
> > >     - The same target distribution $p_\text{tar}(x_0)$
> > >     - The same forward diffusion $p(x_{t+1}|x_t)$ from DDPM sampling
> > >     - Grad parameterization from [3]
> > >
> > > Therefore, when we use Grad parameterization and train from scratch, Train-Grad becomes equivalent to DDS. For the derivation of the training objective, please refer to line 1420 of our revised paper for RLHF with KL regularization, and Proposition 1, equation (21) from [3] for DDS formulation.

---

> ### Author Response · Authors · 2024-11-24
> **Explanation of diffusion sampler behavior**
>
> `I also do not understand why the diffusion sampler (Train-Grad if I understood correctly) captures a mixture of the target and pre-trained distribution, although it was trained from scratch (i.e. has no information of the pre-trained distribution). Or did I misunderstand something in the description of this experiment?`
>
> We sincerely appreciate your careful examination of our paper, as it allows us to address an important technical point about our diffusion sampler.
>
> We acknowledge that our use of "training from scratch" may have caused some confusion. This term refers specifically to the **random initialization** of the diffusion model's parameters, **rather than the absence of pre-trained distribution information**. In fact, DDS **explicitly incorporates the pre-trained distribution in its training objective** through the following KL divergence formulation (see Appendix E for derivation):
> \begin{equation}
> D_{KL}(p_\theta(x_{0:T})||p_\text{tar}(x_{0:T}))
> = D_{KL}(p_\theta(x_{0:T})||p_\text{ref}(x_{0:T})) + \mathbb{E}_{x_0 \sim p_\theta(x_0)}\left[\log \frac{p_\text{tar}(x_0)}{p_\text{ref}(x_0)}\right]
> \end{equation}
> where the target distribution $p_\text{tar}(x_0) \propto p_\text{pre} \exp(r(x_0)/\alpha)$ inherently contains information from the pre-trained distribution.
>
> Regarding the second GMM example, let us clarify the behavior of DDS. While you observed that DDS captures different modes compared to DAS, we want to emphasize that **DDS isn't capturing incorrect modes** - rather, it's **'overfitting' modes that have valid but small probability density** in the target distribution.
>
> To demonstrate this, we can explicitly calculate the target distribution using pre-trained distribution $p_\text{pre}(\mathbb{x}) \propto \sum_{i=1}^8 \exp(-5\Vert \mathbb{x}-(4\cos(i\pi/4), 4\sin(i\pi/4)) \Vert^2/3)$ and reward $r(\mathbb{x}) = -x^2-(y-1)^2/10$ with KL constraint $\alpha=1$ used in the second example:
>
> \begin{align}
> p_\text{tar}(\mathbb{x}) &\propto p_\text{pre} \exp(r(x_0)/\alpha) \propto \sum_{i=1}^8 \exp \bigg( -\frac{8}{3}\left(x -\frac{5}{2}\cos(\frac{i\pi}{4})\right)^2 -\frac{53}{30}\left(y -\frac{200}{53}\sin(\frac{i\pi}{4})-\frac{3}{53}\right)^2 +\frac{50}{3}\cos(\frac{i\pi}{4})^2 + \frac{53}{30}\left(\frac{200}{53}\sin(\frac{i\pi}{4})+\frac{3}{53}\right)^2 \bigg)
> \end{align}
>
> Through this calculation, we can verify that DDS samples come from all 8 modes of p_tar. The density ratio between the highest density mode (i=0) and the third-highest density modes (i=1,3) is approximately exp(4.466) ≈ 87. Considering we only plotted 10,000 samples, the large sample count from these lower-density modes is particularly notable. This indicates that DDS is overemphasizing these modes despite their small probability density, likely due to its direct use of log probability density gradients during training.
>
> We appreciate your attention to this detail and hope this clarification addresses your concern. Please let us know if you have any additional questions.

---

> > ### Comment · Reviewer_QHss · 2024-11-26
> > **Answer to Explanation of diffusion sampler behavior**
> >
> > I sincerely appreciate the authors' comprehensive and detailed response. In the coming days, I will carefully review their answers and determine whether I will raise my evaluation score.
> > Additionally, I would like to provide a technical observation regarding the PDF document. I noticed that the file size is approximately 32 KB, which is significantly larger—up to 10 times the size—compared to other papers of similar page length. This substantial file size results in noticeably longer loading times. I suspect the issue may stem from very high-resolution embedded images.
> > I recommend that the authors investigate the document's image resolution and optimize the file to enhance the overall reading experience.

---

> > > ### Author Response · Authors · 2024-11-28
> > > **Response to Reviewer's Comments and File Size Optimization**
> > >
> > > We sincerely thank the reviewer for their thorough consideration of our responses and the helpful technical observation about the PDF file size. We appreciate your valuable feedback and careful attention to both the technical content and presentation quality throughout this review process. Following your suggestion, we have optimized the document and reduced the file size from 32MB to 4MB while maintaining visual clarity. Please feel free to raise any additional questions or concerns as you review our responses. We greatly value this constructive dialogue and appreciate the time you have devoted to helping us improve our work.

---

> > > ### Author Response · Authors · 2024-12-02
> > > **Gentle Reminder**
> > >
> > > Dear Reviewer,
> > >
> > > I hope this message finds you well. I am writing to kindly follow up on your earlier message regarding your planned review of our manuscript responses.
> > >
> > > Could you please let us know if you have had the opportunity to evaluate our responses?
> > >
> > > Thank you for your time and consideration.
> > >
> > > Best regards

---

> > > > ### Comment · Reviewer_QHss · 2024-12-02
> > > > **Raised Score**
> > > >
> > > > The authors have addressed all of my concerns and I have therefore raised my score to 8.

---

### Official Review · Reviewer_J1vs · 2024-10-29

**Soundness:** 3
**Presentation:** 3
**Contribution:** 3
**Rating:** 8
**Confidence:** 3

**Summary:**

This paper proposes a training-free diffusion sampling method based on Sequential Monte Carlo (SMC) to sample from the reward-aligned target distribution. By incorporating tempering techniques, it offers a robust solution for aligning diffusion models with arbitrary rewards
while preserving general capabilities

**Strengths:**

1. This paper is overall well-written and the motivation is clear. It aims to address the trade-off in diffusion models that align them with specific objectives while maintaining their versatility, which is a critical problem in generative modeling.

2. DAS’s effectiveness is comprehensively validated across diverse scenarios, including toy distribution simulation, single-reward, multi-objective, and online black-box optimization tasks.

**Weaknesses:**

1. More intuitive explanations of SMC are suggested to add between the motivation and method to make it more consistent and intuitive since the introduction of SMC in supplementary material is a bit abstruse to understand, making the superiority of adopting SMC to address the training problem unclear.

2. How to choose hyperparameters such as $\gamma, \alpha$ and particles should be discussed across different scenarios.

**Questions:**

Please see the weaknesses part above.

---

> ### Author Response · Authors · 2024-11-22
>
> We sincerely appreciate the reviewers' constructive comments and positive feedback on our manuscript.
>
> **W1.** `More intuitive explanations of SMC are suggested to add between the motivation and method to make it more consistent and intuitive since the introduction of SMC in supplementary material is a bit abstruse to understand, making the superiority of adopting SMC to address the training problem unclear.`
>
> We sincerely thank the reviewer for their insightful suggestion to improve the intuitiveness of our explanation. To address this concern, we plan to revise the beginning of Section 3.3 in the main text to better clarify the motivation for adopting Sequential Monte Carlo (SMC) methods. Below is our proposed revision:
>
> ---
> Fine-tuning can often lead to over-optimization, while existing methods with approximate guidance struggle to optimize target rewards, making them less attractive as alternatives. To improve approximate guidance during the sampling process, we can use multiple **candidate latents (particles)**. By choosing latents that have both high predicted rewards and remain close to the pre-trained diffusion model at every diffusion step, we can ensure high-quality samples without straying from the original data distribution. This idea aligns closely with the principles of Sequential Monte Carlo (SMC) methods.
>
> SMC works by taking small, guided steps instead of directly sampling from the target distribution. At each step, it evaluates samples (**reweighting**) and focuses on promising ones (**resampling**). When combined with diffusion models, this means generating candidate latents at each step, scoring them using the reward function and the pre-trained model, and resampling only the best candidates. Repeating this process for each step leads to final samples that **align with both the reward and the pre-trained data distribution**.
>
> The challenge is that traditional SMC methods often require hundreds or thousands of particles, which makes them too expensive for computationally heavy diffusion models. To overcome this, we use strategies like **tempering** to improve efficiency. This allows our method to create high-quality, reward-aligned samples at a much lower computational cost, making it **practical for real-world tasks**.
>
> ---
>
> This revision aims to provide an intuitive overview of SMC methods and their integration with our approach, bridging the gap between the motivation and the methodology. By incorporating these changes, we aim to make the rationale for adopting SMC clearer and more accessible while maintaining technical accuracy. We hope this proposed revision addresses the reviewer’s concerns and enhances the overall clarity and coherence of the paper.
>
> **W2.** `How to choose hyperparameters such as` $\gamma$, $\alpha$, `and particles should be discussed across different scenarios.`
>
> Thank you very much for your constructive comments. Our DAS approach mainly involves three hyperparameters: (1) Tempering parameter $\gamma$, (2) Number of particles $N$, and (3) KL coefficient $\alpha$. We use the following guidelines for parameter selection, which is simple and effective in our emprical study.
> - (1) Tempering parameter $\gamma$: Given diffusion step $T$, we set $\gamma$ to satisfy $(1+\gamma)^T \approx 1$ to ensure the final sampled distribution to be $p_{tar}$. In our experiments, we use $\gamma=0.008$ for $T=100$.
> - (2) Number of particles $N$: The smaller number of particles, the faster the sampling. We empirically found that the generated image quality is satisfactory until reducing the number of particles to $N=4$ (See Figure 5 for quanitative and Figure 4 and 11 for qualitative results). Thus, we recommend to set $N$ to be 4.
> - (3) KL coefficient $\alpha$: This should be scaled according to reward magnitude. We use two-level gird search to find $\alpha$ properly and efficiently. For the first level, we perform the grid search for $\alpha \in \{10^{-1}, 10^{-2}, 10^{-3}, 10^{-4}\}$. Then, with the best $\alpha=10^{-k}$ obtained from the first level, we conduct an additional grid search for $\alpha \in \{3 \times 10^{-(k+1)}, 5 \times 10^{-(k+1)}, 10^{-k}, 3 \times 10^{-k}, 5 \times 10^{-k}\}$.
>
> We highlight that the fixed hyperparameters work well across all prompts and the hyperparameter tuning procedure requires **only 8 times of sampling**. We included these guidelines in the revised paper of Appendix D.

---

> > ### Comment · Reviewer_J1vs · 2024-11-25
> >
> > Thanks for the authors' efforts and reply. Since my concern has been addressed, I tend to maintain my score.

---

> ### Author Response · Authors · 2024-11-28
>
> Thank you for your thoughtful review and for confirming that your concerns have been addressed.

---

### Official Review · Reviewer_Kkg4 · 2024-11-04

**Soundness:** 3
**Presentation:** 2
**Contribution:** 2
**Rating:** 5
**Confidence:** 4

**Summary:**

The paper proposes DAS, a training-free approach for aligning diffusion models with specific objectives. It uses Sequential Monte Carlo (SMC) with tempering for reward alignment. This method is demonstrated across generative tasks, including single-reward and multi-objective cases, with performance comparable to fine-tuning methods in terms of target reward optimization and diversity.

**Strengths:**

1. DAS does not require additional training, which reduces computational cost.
2.The use of SMC with tempering is justified through asymptotic properties.
3. DAS balances reward optimization and diversity, and is demonstrated across single-reward, multi-objective, and online settings.

**Weaknesses:**

1. While DAS is compared with fine-tuning and guidance methods, comparisons to baselines like STEGANODE or controlled diffusion could have strengthened the evaluation.
2. DAS assumes differentiable reward functions, which may limit applicability in scenarios involving non-differentiable objectives.
3. Most experiments use Stable Diffusion v1.5, and additional models would have enhanced the generality of the findings.
4. The paper can do more image tasks. Currently it emphasizes findings on aesthetic score, which might not generalize well to other tasks.
5. Limitations: the setup of SMC with tempering, intermediate targets, and backward kernels can be technically demanding. And the effectiveness of DAS depends on the pre-trained model's quality, limiting performance on models with low initial diversity or reward alignment.

**Questions:**

1. The method relies on specific tempering schemes and parameters, and the practical guidelines for selecting these could be more detailed.

---

> ### Author Response · Authors · 2024-11-21
> **Official Comment by Authors (1/3)**
>
> We deeply appreciate the thoughtful review and suggestions for improvement.
>
> **W1.** `While DAS is compared with fine-tuning and guidance methods, comparisons to baselines like STEGANODE or controlled diffusion could have strengthened the evaluation.`
>
> While we are aware of steganography approaches [1,2] and controlled generation methods [3,4] based on diffusion models, we are not exactly sure what method you are referring to by "STEGANODE". Could you kindly specify which works you had in mind when suggesting STEGANODE and controlled diffusion as baselines? This would help us ensure we are addressing your comparison suggestion with the most relevant baselines you are referring to.
>
> While we tried our best to address your questions, we humbly request the specific references you had in mind to help us provide a more targeted response.
>
> ---
>
> [1] Yu, J., Zhang, X., Xu, Y., & Zhang, J. (2024). Cross: Diffusion model makes controllable, robust and secure image steganography. Advances in Neural Information Processing Systems, 36.
> [2] Wei, P., Zhou, Q., Wang, Z., Qian, Z., Zhang, X., & Li, S. (2023). Generative steganography diffusion. arXiv preprint arXiv:2305.03472.
> [3] Zhang, L., Rao, A., & Agrawala, M. (2023). Adding conditional control to text-to-image diffusion models. In Proceedings of the IEEE/CVF International Conference on Computer Vision (pp. 3836-3847).
> [4] Qin, C., Zhang, S., Yu, N., Feng, Y., Yang, X., Zhou, Y., ... & Xu, R. (2023). Unicontrol: A unified diffusion model for controllable visual generation in the wild. arXiv preprint arXiv:2305.11147.
>
> **W2.** `DAS assumes differentiable reward functions, which may limit applicability in scenarios involving non-differentiable objectives.`
>
>
> This is an excellent question. We would like to clarify that DAS _is applicable_ to non-differentiable reward functions by learning compute-efficient surrogate reward functions. In Section 4.3 (lines 518-529), we validated the effectiveness of DAS applied with online fine-tuning methods (UCB and Bootstrap [5]) when a non-differentiable black-box reward function is given. Similarly, DAS can handle any non-diffentiable rewards through an iterative cycle: proposing samples with DAS that maximize the surrogate reward model, recieving feedbacks from the non-differentiable black-box reward, and updating the surrogate reward model to incorporate new reward feedbacks.
>
> Furthermore, we have conducted additional experiments with another non-differentiable function, namely JPEG compressibility, to demonstrate the robustness and effectiveness of DAS. JPEG compressibility is a strict non-differentiable reward function measured by the negative file size (in KB) after JPEG compression at quality factor 95.
>
> **Configuration.** We used the same configuration with experiments in Section 4.3 except using 4096 reward queries and KL coefficient $\alpha=0.1$.
>
> **Result.** Reward values of the generated images are presented in the table below. Our method achieves the best compressibility score, outperforming both pre-trained model and DDPO, which can naturally incorporate non-differentiable rewards using RL. Also, it maintains CLIPScore and diversity metrics (TCE and LPIPS_MPD) compared to pre-trained model, which implies over-optimization is mitigated as intended.
>
> |               | SD1.5    | DDPO  | DAS-UCB  |
> | -------------------- | -------- | ----- | -------- |
> | JPEG Compressibility | -116      | -84    | **-71**       |
> | CLIPScore            | **0.260**    | 0.258 | 0.258    |
> | TCE                  | 39.8 | 41.1     | **43.4** |
> | LPIPS_MPD            | **0.660**  | 0.624     | 0.618  |
>
> Visualizations of the samples are provided in Appendix F.3 which show how our method effectively minimizes background complexity while preserving key semantic features of the subjects.
>
> In summary, while DAS is designed for differentiable rewards, our black-box optimization approach provides an effective solution for non-differentiable objectives. We have clarified the methodology and included detailed experimental results in the revised manuscript (Appendix F.3).
>
> ---
> [5] Uehara, M., Zhao, Y., Black, K., Hajiramezanali, E., Scalia, G., Diamant, N. L., ... & Biancalani, T. Feedback Efficient Online Fine-Tuning of Diffusion Models. In Forty-first International Conference on Machine Learning.

---

> ### Author Response · Authors · 2024-11-21
> **Official Comment by Authors (2/3)**
>
> **W3.** `Most experiments use Stable Diffusion v1.5, and additional models would have enhanced the generality of the findings.`
>
> Thank you very much to help us improve our manuscript. As per your suggestion, we have extended our experiments using SDXL to demonstrate the generality of our approach across the diffusion backbones. As shown in the table below, **DAS using SDXL as a backbone still outperforms the baselines**, including the base SDXL [6] (with refiner) and DPO-SDXL, which fine-tuned SDXL using DiffusionDPO [7], achieving superior performance in both target optimization (PickScore) and cross-reward generalization (HPSv2, ImageReward) while maintaining competitive diversity metrics (TCE, LPIPS_MPD) in most cases even with just 4 particles.
>
> |                    | SDXL (base + refiner) | DPO-SDXL     | DAS-SDXl (Ours) |
> | ------------------ | --------------------- | ------------ | --------------- |
> | PickScore (target) | 0.225               | 0.232      | **0.255**     |
> | HPSv2              | 0.285              | 0.294      | **0.306**     |
> | ImageReward        | 0.82               | 1.24      | **1.40**     |
> | TCE                | 46.1              | **46.7** | 46.1        |
> | LPIPS_MPD          | 0.567               | 0.606      | **0.608**     |
>
> Qualitative comparisons of **generated images are provided in Figure 11 of Appendix F.2**, which show high-quality and improved text-image alignment results of our method beyound baselines.
>
> ---
>
> [6] Dustin Podell, Zion English, Kyle Lacey, Andreas Blattmann, Tim Dockhorn, Jonas M¨uller, Joe Penna, and Robin Rombach. Sdxl: Improving latent diffusion models for high-resolution image synthesis. In The Twelfth International Conference on Learning Representations, 2024.
> [7] Bram Wallace, Meihua Dang, Rafael Rafailov, Linqi Zhou, Aaron Lou, Senthil Purushwalkam, Stefano Ermon, Caiming Xiong, Shafiq Joty, and Nikhil Naik. Diffusion Model Alignment Using Direct Preference Optimization. In 2024 IEEE/CVF Conference on Computer Vision and Pattern Recognition (CVPR)
>
> **W4.** `The paper can do more image tasks. Currently it emphasizes findings on aesthetic score, which might not generalize well to other tasks.`
>
> We appreciate the reviewer's suggestion for expanding our experimental scope. We would first like to respectfully note that our paper already **demonstrated DAS's effectiveness across several diverse objectives beyond aesthetic score** including PickScore which represent human preference (Section 4.1) and CLIPScore which represent text-image alignment (Section 4.2) apart from aesthetic score. Even though Section 4.2 focused on multiple rewards setting, Figure 6(a) shows that DAS significantly outperforms DDPO and AlignProp when optimizing CLIPScore alone, too. However, there are indeed more various image tasks, and we added JPEG compressibility to address your concern in Q2 and Q4 (detail in the response to Q2).
>
> These diverse experiments demonstrate that DAS's benefits generalize well across different types of objectives beyond aesthetic quality. We revised the paper to better highlight this variety of tasks and their distinct characteristics.

---

> ### Author Response · Authors · 2024-11-21
> **Official Comment by Authors (3/3)**
>
> **W5.**
>
> 1) `Limitations: the setup of SMC with tempering, intermediate targets, and backward kernels can be technically demanding.`
>
> We acknowledge that implementing SMC methods may present initial technical challenges.  However, we have developed our implementation based on the widely-used diffusers library (See diffusers_patch.pipeline_using_SMC in the supplementary codes), enabling users to implement DAS with just a few lines of code. We will open-source our code after publication, and we hope this will help lower the technical barrier of DAS and contribute to the AI community.
>
> 2) `And the effectiveness of DAS depends on the pre-trained model's quality, limiting performance on models with low initial diversity or reward alignment.`
>
> To check if the effectiveness of DAS depends on the pre-trained model's quality, we conducted additional experiments that compares the performance of DAS when combined with SD1.5 and SDXL. As shown in the table below, DAS improves the target reward score (PickScore) and cross reward scores (HPSv2 and ImageReward) while maintaining the diversity metrics (TCE and LPIPS_MPD) regardless of the backbone's quality. Specifically, looking at the target reward (PickScore), DAS-SD1.5 achieves the most performance improvements, outpeforming the vanilla SDXL and performing nearly identical to DAS-SDXL. This indicates that the effectiveness of DAS does not depend on the quality of pre-trained models.
>
> | Metric | SD1.5 | DAS-SD1.5 | SDXL | DAS-SDXl |
> | ------------------ | ------- | ----- | ---------------- | -------------- |
> | PickScore (target) | 0.220 | **0.256 (+0.036)** | 0.225 | **0.255 (+0.031)** |
> | HPSv2              | 0.284 | **0.303 (+0.019)** | 0.285 | **0.306 (+0.021)** |
> | ImageReward        | 0.41  | **1.06 (+0.65)**   | 0.82  | **1.40 (+0.58)**   |
> | TCE                | 46.0  | **46.0**           | 46.1 | **46.1**        |
> | LPIPS_MPD          | 0.662 | 0.647              | 0.568 | **0.608**      |
>
> Therefore, we conclude that **DAS is effective regardless of the initial model quality**, and we will include this finding in the final manuscript.
>
> **Q1.** `The method relies on specific tempering schemes and parameters, and the practical guidelines for selecting these could be more detailed.`
>
> Thank you very much for your constructive comments. Our DAS approach mainly involves three hyperparameters: (1) Tempering parameter $\gamma$, (2) Number of particles $N$, and (3) KL coefficient $\alpha$. We use the following guidelines for parameter selection, which is simple and effective in our emprical study.
> - (1) Tempering parameter $\gamma$: Given diffusion step $T$, we set $\gamma$ to satisfy $(1+\gamma)^T \approx 1$ to ensure the final sampled distribution to be $p_{tar}$. In our experiments, we use $\gamma=0.008$ for $T=100$.
> - (2) Number of particles $N$: The smaller number of particles, the faster the sampling. We empirically found that the generated image quality is satisfactory until reducing the number of particles to $N=4$ (See Figure 5 for quanitative and Figure 4 and 11 for qualitative results). Thus, we recommend to set $N$ to be 4.
> - (3) KL coefficient $\alpha$: This should be scaled according to reward magnitude. We use two-level gird search to find $\alpha$ properly and efficiently. For the first level, we perform the grid search for $\alpha \in \{10^{-1}, 10^{-2}, 10^{-3}, 10^{-4}\}$. Then, with the best $\alpha=10^{-k}$ obtained from the first level, we conduct an additional grid search for $\alpha \in \{3 \times 10^{-(k+1)}, 5 \times 10^{-(k+1)}, 10^{-k}, 3 \times 10^{-k}, 5 \times 10^{-k}\}$.
>
> We highlight that the fixed hyperparameters work well across all prompts and the hyperparameter tuning procedure requires **only 8 times of sampling**. We included these guidelines in the revised paper of Appendix D.

---

> ### Author Response · Authors · 2024-11-24
> **Gentle Reminder**
>
> Dear Reviewer,
>
> I hope you are doing well. We submitted our rebuttal addressing your thoughtful comments on November 22nd. Additionally, we have updated our paper to incorporate these changes, specifically:
>
> - Line 528 and Appendix F.3 regarding online black-box optimization for non-differentiable rewards
> - Appendix F.2 presenting new experiments using the latest SDXL as backbone
> - Lines 1400 ~ 1409 in Appendix D regarding hyperparameter selection
> - Line 321 clarifying our task definition about single reward task
>
> We believe we have thoroughly addressed your concerns both in our response and in the revised paper, and would greatly value your feedback. Please let us know if any points require further clarification.
>
> Thank you for your time and consideration.
>
> Best regards

---

> ### Author Response · Authors · 2024-11-27
> **Regarding W1**
>
> **W1.** `While DAS is compared with fine-tuning and guidance methods, comparisons to baselines like STEGANODE or controlled diffusion could have strengthened the evaluation.`
>
> Regarding W1, while we await clarification, we have carefully considered your suggestion and would like to address why these methods might **not be directly applicable to our problem setting**. Below, we outline the differences between these approaches and our problem setting for DAS to clarify the evaluation framework:
>
> **1. Task.** The fundamental objectives differ significantly. Steganography methods aim to hide information within generated images, while controlled diffusion methods generate images based on specific conditions (edges, depth, pose, etc.). In contrast, our work addresses the broader challenge of general reward maximization through diffusion models.
>
> **2. Dataset and Training Objective.** The key technical distinction lies in training requirements:
>
> - Steganography/controlled diffusion methods **utilize data from their target distribution, enabling score matching loss** from conventional diffusion model training
> - Our reward maximization setting **cannot assume access to high-reward samples (i.e. target distribution samples), making score matching infeasible**
> - This fundamental limitation explains why existing approaches use alternatives like RL or direct backpropagation, which lead to the over-optimization challenges we address
> - Our training-free solution specifically tackles these limitations
>
> We hope this clarifies the methodological gaps between the suggested baselines and our problem setting. We welcome additional discussion once you specify the particular references you had in mind. If you have any feedback on our previous rebuttal, we would be happy to address those points as well.

---

> > ### Comment · Reviewer_Kkg4 · 2024-12-02
> > **Response to rebuttal**
> >
> > I appreciate the rebuttal from the authors. However, I believe more baselines can strengthen the paper. Other than the works mentioned by the authors, works [1-7] are related to the task of reward maximization for diffusion model and should be considered. Thus I keep my score for now.
> >
> > [1] Diffusion-Stego: Training-free Diffusion Generative Steganography via Message Projection
> >
> > [2] LDStega: Practical and Robust Generative Image Steganography based on Latent Diffusion Models
> >
> > [3] Amortizing intractable inference in diffusion models for vision, language, and control
> >
> > [4] Steering Masked Discrete Diffusion Models via Discrete Denoising Posterior Prediction
> >
> > [5] Practical and asymptotically exact conditional sampling in diffusion models
> >
> > [6] Probabilistic Inference in Language Models via Twisted Sequential Monte Carlo
> >
> > [7] Design-Bench: Benchmarks for Data-Driven Offline Model-Based Optimization

---

> > > ### Author Response · Authors · 2024-12-03
> > > **Response to Baseline Suggestion**
> > >
> > > We thank the reviewer for these suggestions. After carefully analyzing each work, we found that most focus on different problem settings, and others have limitations compared to our approach.
> > >
> > > **Works with different problem settings:**
> > >
> > > - [1,2] are designed specifically for steganography, which aims to embed hidden messages during image generation. Unlike our goal of aligning diffusion models with general reward functions, steganography methods focus on encoding and decoding specific information within the generation process, making the approaches fundamentally incompatible.
> > > - [4] targets discrete diffusion models, making it incompatible with continuous diffusion model alignment
> > > - [6] targets autoregressive language models, which condition each token generation on all previous tokens in a left-to-right manner in discrete token space. In contrast, continuous diffusion models use Markov transitions that only depend on the previous latent in continuous latent space. This fundamental difference in both dependency structure and sampling space makes their approach unsuitable for our setting, despite also using SMC methods.
> > > - [7] addresses broad model-based optimization rather than diffusion model enhancement
> > >
> > > **Works with limitations:**
> > >
> > > - [3] requires **computationally expensive fine-tuning** and exhibits **diversity reduction** (Figure 3, Table H.1). Most critically, it has only been validated on **single prompts** for T2I models, raising concerns about generalization to unseen prompts
> > > - [5] only focuses on **class-conditional** generation and inpainting tasks, which are easier than complex reward-maximiation. Also, it just applies SMC **without incorporating advanced technique like tempering**. Our ablation study (Figure 5) demonstrates that tempering is crucial for reward-maximiation.
> > >
> > > **Our novelty**
> > > - Achieving superior performance (e.g., reward-maximization, preserving diversity, and sample efficiency) by introducing the tempering technique
> > > - Handling diverse reward-maximiation scenarios
> > >
> > > We will include this discussion in the final manuscript to better position our contributions. Given that the discussion period is drawing to a close, we would be grateful for any updated feedback and remain available to address any additional questions or concerns you may have.

---

### Author Response · Authors · 2024-11-21
**General Response**

We sincerely appreciate the reviewers' positive feedback and valuable comments. Most reviewers agreed that (1) **the problem setting and motivation are clear** (Reviewer J1vs, QHss and 31k3), (2) **the methodology is sound and justified by theoretic analysis** (Reviewer Kkg4, J1vs and 31k3), (3) **the evaluation was performed extensively on diverse reward-alignment scenarios** (Reviewer Kkg4, J1vs, and 31k3), and (4) **the presentation is clear** (Reviewer J1vs and 31k3). During the rebuttal, we addressed the reviewers' remaining concerns by providing clarification with comprehensive additional experimental results (by adding the latest diffusion backbones, three baselines, one non-differentiable reward case, and a more challenging synthetic dataset; See the attached PDF file for visual results). We hope that the remaining concerns are successfully addressed by the rebuttal and are happy to answer more questions during the discussion period.

---

### Author Response · Authors · 2024-11-28
**Final Revision**

We sincerely thank all reviewers for their constructive feedback that helped strengthen our work. The final revised paper incorporates several major improvements:

1. Enhanced Accessibility and Understanding
- Intuitive explanation of DAS methodology (Section 3.3 lines 196 ~ 207)
- Clear hyperparameter selection guidelines (Appendix D)
- Thorough comparison with diffusion-based samplers (Appendix E)
- Additional Swiss roll visualization (Appendix F.1)


2. Extended Applications
- Testing with SDXL architecture (Appendix F.2)
- Non-differentiable rewards via online black-box optimization (Section 4.3, Appendix F.3)
- Discussion of future works (Appendix G)


3. Improved Readability
- Optimized file size (<4MB)
- Improved notation timing and definitions in Section 3
- Enhanced evaluation metrics explanation (Section 4.1)
- Clarified optimality variable formulation (Section 3.1)

We believe these revisions address the reviewers' concerns while making the paper more accessible and practically useful. For easy reference, all changes in the revised manuscript are highlighted in green.

Again, we appreciate the thorough review process that helped improve the quality of our work.

---

### Meta-Review · Area_Chair_6wm1 · 2024-12-19

**Metareview:**

The paper considers aligning (tilting) the diffusion model with reward. Formally, this corresponds to sampling from the following product of densities
$$p_{\text{target}} \propto p_{\text{pretrained}}(x)\exp(\beta r(x)),$$
where $p_{\text{pretrained}}(x)$ is the density of samples produced by the pre-trained diffusion model, $r(x)$ is a predefined reward model, and $\beta$ is the inverse temperature hyperparameter.

To sample from this product, the authors employ the Sequential Monte Carlo (SMC) framework analogously align (tilt) the pretrained marginals along the way by multiplying it with $\exp(\beta_t r(x))$. Furthermore, the authors introduce the transition kernel that minimizes the variance of the SMC weights (necessary for correct sampling), which results in a very practical scheme which is demonstrated empirically.

The reviewers unanimously recognize both the practical and theoretical contributions of the paper, which leaves no doubts about its acceptance.

**Additional Comments On Reviewer Discussion:**

Most of the concerns raised by the reviewers were regarding the additional evaluations of the proposed algorithm. These concerns were successfully addressed during rebuttal by providing new experimental results.

---

### Decision · Program_Chairs · 2025-01-22

Accept (Spotlight)